# Conserved allosteric inhibition mechanism in SLC1 transporters

Yang Dong[1], Jiali Wang[1], Rachel-Ann Garibsingh[2], Keino Hutchinson[2], Yueyue Shi[1], Gilad Eisenberg[1], Xiaozhen Yu[1], Avner Schlessinger[2]*, Christof Grewer[1]*

[1]Department of Chemistry, Binghamton University, Binghamton, United States; [2]Department of Pharmacological Sciences, Icahn School of Medicine at Mount Sinai, New York, United States

**Abstract** Excitatory amino acid transporter 1 (EAAT1) is a glutamate transporter belonging to the SLC1 family of solute carriers. It plays a key role in the regulation of the extracellular glutamate concentration in the mammalian brain. The structure of EAAT1 was determined in complex with UCPH-101, apotent, non-competitive inhibitor of EAAT1. Alanine serine cysteine transporter 2 (ASCT2) is a neutral amino acid transporter, which regulates pools of amino acids such as glutamine between intracellular and extracellular compartments . ASCT2 also belongs to the SLC1 family and shares 58% sequence similarity with EAAT1. However, allosteric modulation of ASCT2 via non-competitive inhibitors is unknown. Here, we explore the UCPH-101 inhibitory mechanisms of EAAT1 and ASCT2 by using rapid kinetic experiments. Our results show that UCPH-101 slows substrate translocation rather than substrate or $Na^+$ binding, confirming a non-competitive inhibitory mechanism, but only partially inhibits wild-type ASCT2. Guided by computational modeling using ligand docking and molecular dynamics simulations, we selected two residues involved in UCPH-101/EAAT1 interaction, which were mutated in ASCT2 (F136Y, I237M, F136Y/I237M) in the corresponding positions. We show that in the F136Y/I237M double-mutant transporter, 100% of the inhibitory effect of UCPH-101 could be restored, and the apparent affinity was increased ($K_i = 4.3\ \mu M$), much closer to the EAAT1 value of 0.6 μM. Finally, we identify a novel non-competitive ASCT2 inhibitor, through virtual screening and experimental testing against the allosteric site, further supporting its localization. Together, these data indicate that the mechanism of allosteric modulation is conserved between EAAT1 and ASCT2. Due to the difference in binding site residues between ASCT2 and EAAT1, these results raise the possibility that more potent, and potentially selective ASCT2 allosteric inhibitors can be designed .

*For correspondence:
avner.schlessinger@mssm.edu
(AS);
cgrewer@binghamton.edu (CG)

Competing interest: The authors declare that no competing interests exist.

## Editor's evaluation

The goal of this study is to identify allosteric modulators of an SLC-1 amino acid transporter, ASCT2, which has been implicated in cancer progression. By combining computational and docking methods with functional measurements, this study provides convincing evidence for a conserved allosteric SLC-1 inhibition mechanism. The findings are important to the fields of transporter mechanism and SLC-1 pharmacology.

## Introduction

Excitatory amino acid transporters (EAATs) from the solute carrier 1 (SLC1) family are an important class of membrane proteins, which are strongly expressed in the mammalian central nervous system (CNS). EAATs are responsible for the transport of glutamate across neuronal and astrocytic membranes in the CNS. Glutamate transporters are secondary active transporters, which utilize energy from the

sodium concentration gradient between the extra- and intracellular sides of the membrane to take up glutamate into cells against a large concentration gradient (*Wadiche et al., 1995*; *Zerangue and Kavanaugh, 1996a*). In addition, H⁺ is co-transported and K⁺ is counter-transported (*Zerangue and Kavanaugh, 1996a*). This glutamate uptake is an important process, preventing glutamate concentrations from reaching neurotoxic levels in the extracellular space (*Zerangue and Kavanaugh, 1996a*; *Tanaka et al., 1997*).

The SLC1 family contains five glutamate transporters (EAATs1–5), as well as two neutral amino acid transporters, the alanine serine cysteine transporters alanine serine cysteine transporter 1 and 2 (ASCT1 and -2) (*Arriza et al., 1993*; *Utsunomiya-Tate et al., 1996*; *Zerangue and Kavanaugh, 1996b*). ASCT2 has been implicated in rapidly growing cancer cells as a supply mechanism for glutamine, and inhibition of the transporter was shown to reduce cell proliferation and tumor size (*Bröer et al., 1999*; *Son et al., 2013*; *Wahi and Holst, 2019*; *Kanai et al., 2013*). Therefore, ASCT2 is emerging as a promising drug target.

Structurally, SLC1 family members are assembled from three identical subunits (*Canul-Tec et al., 2017*; *Garaeva et al., 2019*; *Yu et al., 2019*), each contain eight transmembrane domains (TM1–8). TMs 1, 2, 4, and 5 form the trimerization domain, and TMs 3, 6, 7, and 8, together with hairpin loops HP1 and HP2, form the transport domain, which is based on an inverted repeat structure (*Crisman et al., 2009*; *Reyes et al., 2009*). Glutamate transporter inhibitors were widely studied for pharmacological purposes. Several types of competitive inhibitors were developed based on an aspartate scaffold, for example, TBOA (DL-*threo*-beta-benzyloxyaspartate) (*Shimamoto et al., 1998*) and TFB-TBOA ((3S)-3-[[3-[[4-(trifluoromethyl)benzoyl]amino]phenyl]methoxy]-L-aspartic acid) (*Shimamoto et al., 2004*). In addition, non-competitive inhibitors were identified such as UCPH-101 and -102 (*Erichsen et al., 2010*; *Abrahamsen et al., 2013*). Characterization with electrophysiological methods suggested UCPH-101 was specific for EAAT1 rather than EAATs 2–5. The inhibition of EAAT1 by UCPH-101 was not affected by glutamate or competitive inhibitors, such as TBOA (*Shimamoto et al., 1998*). Subsequently, the binding site of UCPH-101 was identified, first by site-directed mutagenesis, and then in a UCPH-101-bound crystal structure (*Canul-Tec et al., 2017*). UCPH-101 was found to bind to a hydrophobic region between the trimerization and transport domains (*Canul-Tec et al., 2017*).

In addition to UCPH-101, other allosteric modulators, GT949 and derivatives, were first introduced by Falucci et al. (*Falcucci et al., 2019*; *Kortagere et al., 2018*). GT949 is a positive allosteric modulator in EAAT2. However, it operates as an inhibitor in other EAAT subtypes. After incubation of GT949, glutamate uptake activity and $v_{max}$ increase in EAAT2, without altering the substrate affinity. The report also suggested that the allosteric binding site is located between the transport and the trimerization domains (*Falcucci et al., 2019*; *Kortagere et al., 2018*).

For ASCTs, inhibitor pharmacology is less well studied. Currently, a number of competitive inhibitors have been identified, on the basis of structural similarity with EAAT inhibitors, as well as docking to homology models (*Singh et al., 2017*; *Ndaru et al., 2019*). While these newly discovered substrate binding site inhibitors show improved potencies and selectivities (*Garibsingh et al., 2021*), because of their high similarity to amino acid substrates they (i) likely have poor bioavailability and (ii) need to compete with the amino acid-rich media; thus, they are unlikely to be useful as future anti-cancer drugs via inhibition of ASCT2-mediated transport. It is therefore critical to identify specific, allosteric, non-amino acid-like inhibitors for ASCT2. EAAT1 and ASCT2 share 57.6% sequence similarity and 38.5% sequence identity (*Figure 1—figure supplement 1*); however, it is not known if the allosteric binding site of EAAT1 is also found in ASCTs, and whether small molecules can modulate ASCT2 non-competitively via this potential site. Structural analysis of the allosteric sites of EAAT1 and ASCT2 suggests that two non-conserved residues in EAAT1 (e.g., F136, and I237) significantly contribute to the UCPH-101 interaction, while other binding site residues that are less critical for binding are conserved between EAAT1 and ASCT2 (*Figure 1—figure supplement 1*).

In this report, we first compare and detail the effects of UCPH-101 on the SLC1 family members EAAT1 and ASCT2, including analysis of rapid chemical kinetic experiments. While UCPH-101 was found to be a partial and low affinity inhibitor of wild-type ASCT2, engineering of a double-mutant ASCT2 transporter based on computational docking analysis on the basis of the EAAT1 binding site resulted in a largely recovered UCPH-101 effect. In addition, we describe the identification of a non-competitive inhibitor that is not related to UCPH-101 from a virtual screen. Finally, we discuss the importance of our results to the understanding of allosteric inhibition in SLC1 transporters as well as

their relevance to the identification of future, non-competitive modulators for ASCTs, including anti-cancer drugs.

## Results

UCPH-101 has been previously reported to be a potent non-competitive glutamate transporter inhibitor with high specificity for EAAT1 rather than the other SLC1 glutamate transporter subtypes. The EAAT1 structure shows UCPH-101 (*Figure 1A*) deeply buried between the transport domain and the trimerization domain interface (*Figure 1B and C*). The hydrophobic pocket is located between TM3, TM7, and TM4 (*Figure 1C*). The residues that interact with UCPH-101 include G120 (TM3), V373 (TM7a), M231 (TM4c), Y127 (TM3), F369 (TM7), and F235 (TM4c) (*Figure 1D*). This binding pocket is at a distance of over 15 Å from the substrate binding site (*Figure 1B*), suggesting that UCPH-101 may not preclude substrate binding (*Canul-Tec et al., 2017*).

A comparison of the UCPH-101 binding pocket between EAAT1 and ASCT2 (*Figures 1C and 2B and C*) reveals similarities and differences in the overall shape and physical chemical properties of the allosteric binding site of EAAT1 and the equivalent region in ASCT2 (*Figures 1D and 2C*). Several residues are conserved between these proteins maintaining the overall shape of the site. For example, L104, V227, F369 in EAAT1 correspond to L112, V235, and F377 in ASCT2. Conversely, some residues are not fully conserved. Most notably, Y127 and M271 correspond to F136 and I237 in ASCT2, affecting the shape and polarity of the binding site (*Figure 1—figure supplement 1* and *Figure 2B and C*). We hypothesized that these differences as well as other changes will lead to differences in inhibitor specificity of the corresponding allosteric sites.

Indeed, molecular docking of UCPH-101 to the putative allosteric site of ASCT2 fails using multiple approaches including both rigid and flexible docking, as well as constraints on different interactions and residues (Materials and methods). However, when F136 and I273 are remodeled to tyrosine and methionine respectively, UCPH-101 can be successfully docked into ASCT2 with a pose similar to that seen in the EAAT1 structures (see below). This result added further support to our hypothesis that Y127 and M271 are key residues impacting binding of UCPH-101 in EAAT1 and explains why inhibition is only partial in wild-type ASCT2 (see below). Next, our goal was to experimentally test predictions from docking with respect to interaction of UCPH-101 with the SLC1 family member ASCT2, contrasting them with results from EAAT1. To validate our experimental system and to compare inhibitor effects on pre-steady-state currents, which had not been investigated in the past, we first briefly summarize UCPH-101 effects on EAAT1.

## UCPH-101 is a slow-binding, non-competitive inhibitor of EAAT1, which blocks translocation rather than other partial reactions of the transport cycle

As reported previously (*Erichsen et al., 2010*; *Abrahamsen et al., 2013*; *Jensen et al., 2009*), glutamate-induced EAAT1 anion current was inhibited in a concentration-dependent manner by UCPH-101 (*Figure 3A and B*). At low [UCPH-101] (2 µM) (*Figure 3A*, red trace) onset of inhibition was slow with complete inhibition reached only after several seconds. In agreement with the expectation of a non-competitive inhibition mechanism, the apparent $K_i$ was essentially independent of the glutamate concentration (*Figure 3B and C*). In contrast, if UCPH-101 would show a competitive inhibition pattern, an increase of apparent $K_i$ with the glutamate concentration would be expected, according to *Equation 2*.

$$K_i(S) = K_i(0) + [S]K_i(0)/K_m(substrate) \tag{1}$$

In *Equation 1*, $K_i(0)$ and $K_i(S)$ are the $K_i$ values (inhibition constants) in absence and presence of glutamate. $K_m(substrate)$ is the apparent Michaelis-Menten constant for glutamate, and $[S]$ is the substrate concentration. Furthermore, we determined the voltage dependence of the UCPH-101 inhibitory effect. As expected, anion currents were fully inhibited at all membrane potentials (*Figure 3—figure supplement 1*).

To obtain a high time resolution for determining the effect of UCPH-101 on transport kinetics, we applied glutamate rapidly to EAAT1 using a piezo-based solution exchange device. This system has a 5–10 ms time resolution when applied to whole cells, and also allows the rapid removal of

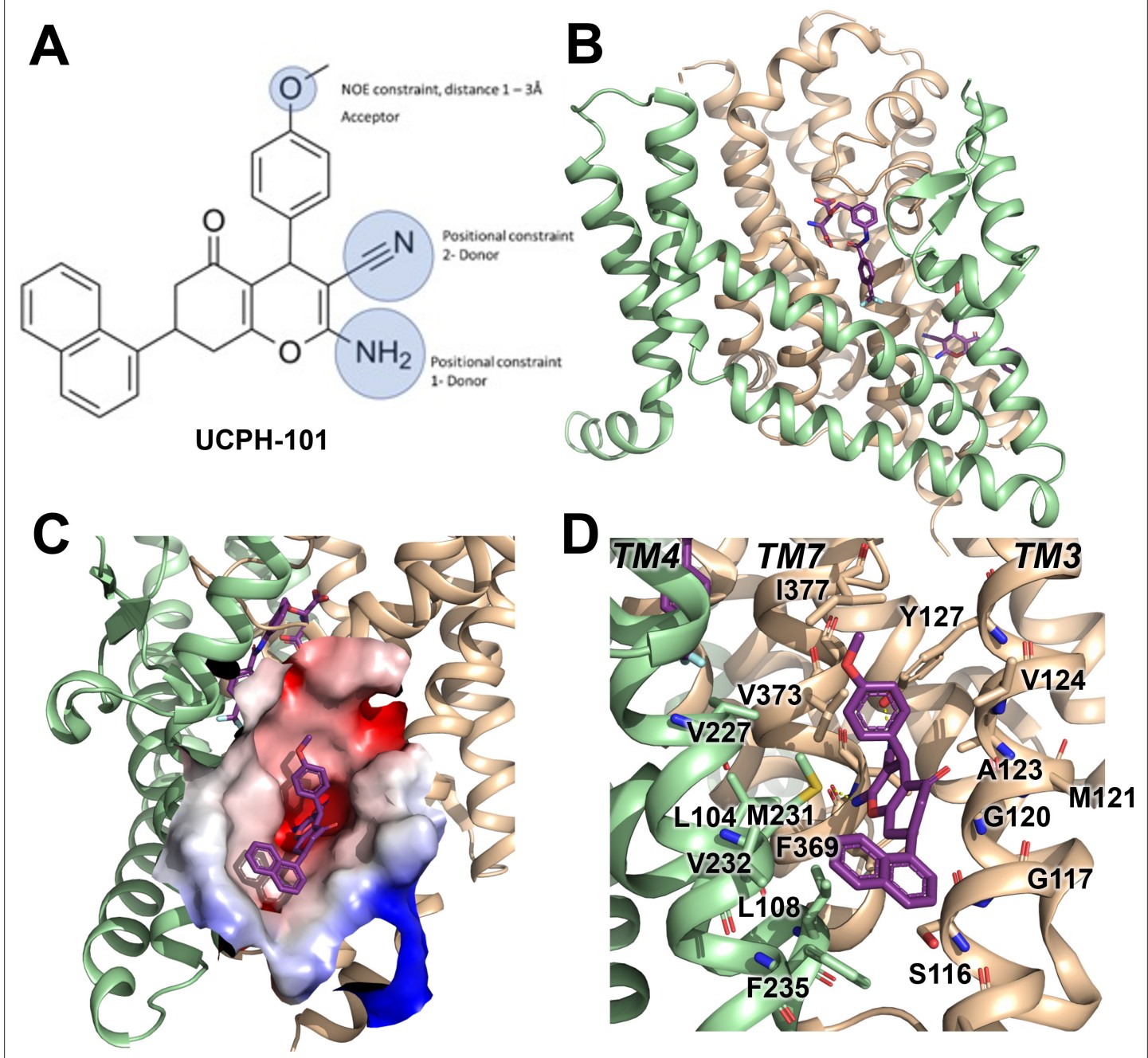

**Figure 1.** Structure of the excitatory amino acid transporter 1 (EAAT1) UCPH-101-bound state. (**A**) UCPH-101 structure and substituents used for constraints in docking calculations. (**B**) EAAT1 structure (PDB id: 5MJU) in complex with the competitive inhibitor TFB-TBOA ((3S)-3-[[3-[[4-(trifluoromethyl)benzoyl]amino]phenyl]methoxy]-L-aspartic acid) (cyan sticks) in the substrate binding site and the allosteric inhibitor UCPH-101 (purple sticks) in the allosteric site. The trimerization and transport domain are highlighted in light green and orange, respectively. (**C**) Illustration of the UCPH-101 binding site at the domain interface. The electrostatic surface calculated using the Adaptive Poisson-Boltzmann Solver (APBS) plugin in Pymol (default parameters) in the absence of UCPH-101 is highlighted. (**D**) EAAT1 transmembrane (TM) helices and amino acid residues in proximity to the UCPH-101 binding site.

The online version of this article includes the following figure supplement(s) for figure 1:

**Figure supplement 1.** EAAT1/ASCT2/ASCT1 sequence alignment.

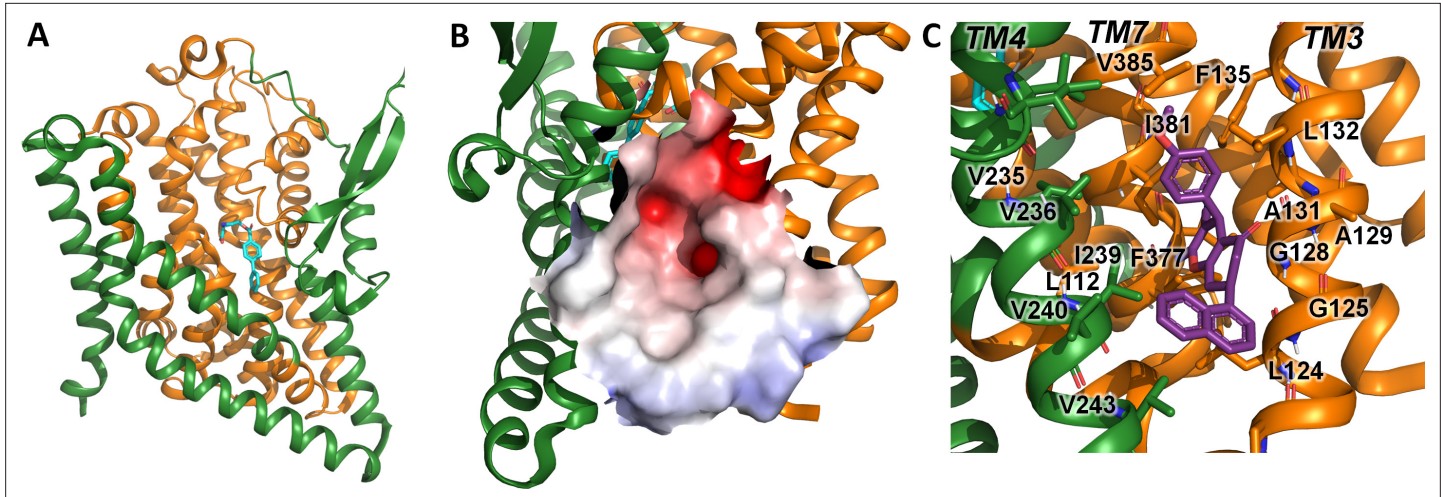

**Figure 2.** Alanine serine cysteine transporter 2 (ASCT2) structures and predicted allosteric binding site. (**A**) ASCT2 structure (PDB id 7BCS) with the competitive inhibitor *L-cis*-PBE (cyan sticks). The trimerization and transport domains are highlighted in dark green and dark orange, respectively. (**B**) Illustration of the putative allosteric binding site (UCPH-101 binding site in excitatory amino acid transporter 1 [EAAT1]) at the domain interface. The electrostatic surface of the binding site is highlighted. (**C**) Close-up view of the putative ASCT2 allosteric binding site; UCPH-101 coordinates (purple sticks) were derived from the EAAT1 structure in the corresponding site.

substrate, providing information on the recovery of current after glutamate removal. However, due to the millisecond time resolution, it will limit determination of very early intermediates in the transport cycle, which are present in a sub-millisecond time domain, but the time resolution is high enough for determining glutamate transporter turnover rate, and how it is affected by the presence of inhibitor. EAAT1 anion currents (*Figure 4A*) and transport currents (*Figure 4E*) show rapid rise of the current after glutamate application within the time resolution afforded by solution exchange (5–10 ms). The rise of the current is followed by rapid decay of a transient current component to the steady state, as demonstrated previously (*Grewer et al., 2000*). This transient phase of the current was previously

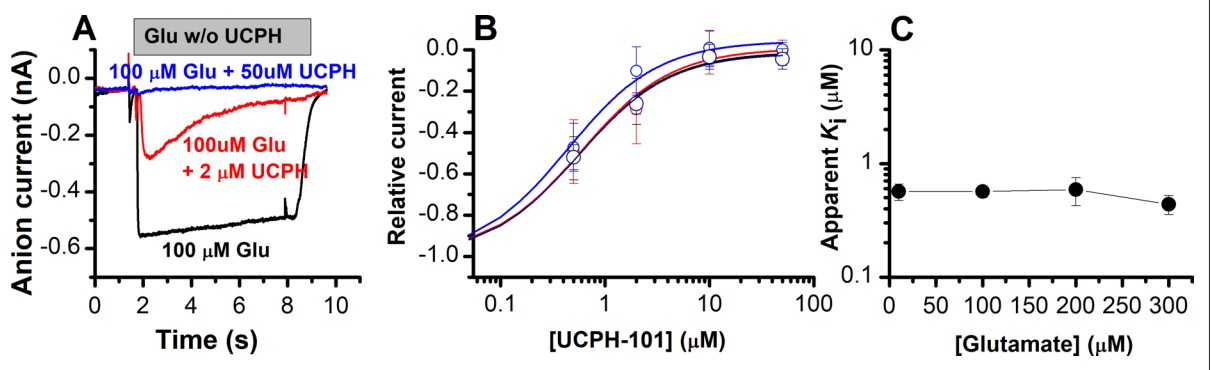

**Figure 3.** UCPH-101 is a high-affinity, non-competitive inhibitor of excitatory amino acid transporter 1 (EAAT1) anion current. (**A**) Typical whole-cell current recordings in the absence (black) and presence of 2 μM (red) and 100 μM (blue) UCPH-101. [Glutamate] was 100 μM. Experiments were performed using 140 mM sodium methanesulfonate (NaMes) in the extracellular buffer, and 130 mM KSCN intracellularly (forward transport conditions). (**B**) Dose-response curves to determine the apparent $K_i$ for UCPH-101 at increasing glutamate concentration of 100 μM (black, *n*=16), 200 μM (red, *n*=19), and 300 μM (blue, *n*=13). (**C**) Glutamate concentration dependence of the apparent $K_i$ for UCPH-101 suggests non-competitive inhibition mechanism. All experiments were performed at 0 mV membrane potential.

The online version of this article includes the following source data and figure supplement(s) for figure 3:

**Source data 1.** The source data contains original current traces for *Figure 3A*, and the original data for panels (B) and (C).

**Figure supplement 1.** Excitatory amino acid transporter 1 (EAAT1) anion current is inhibited by UCPH-101 independent of voltage.

**Figure supplement 1—source data 1.** The source data contains the current traces for *Figure 3—figure supplement 1A and B*, and the original data for panel (C).

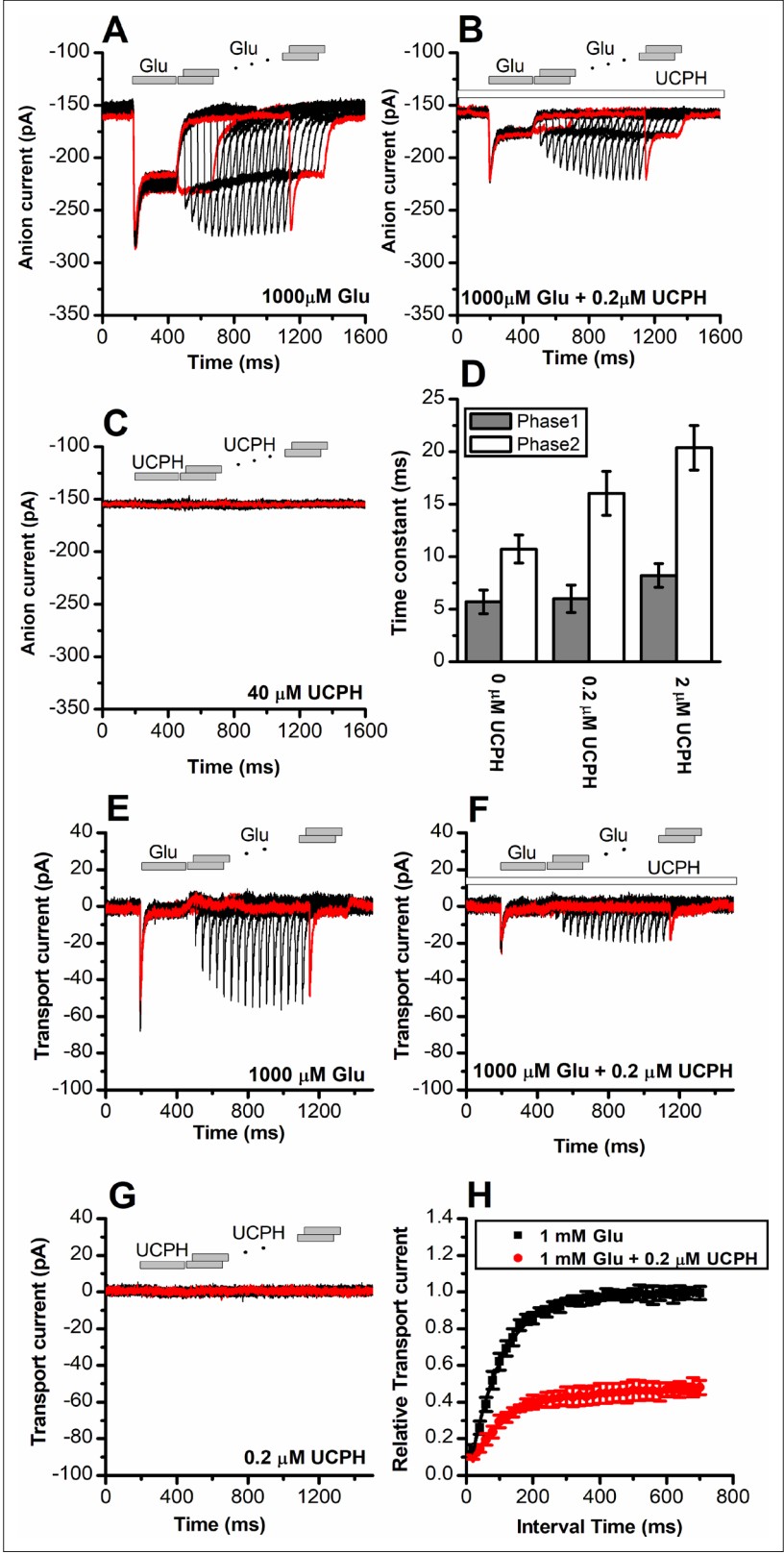

**Figure 4.** UCPH-101 has only minor effect on recovery kinetics of excitatory amino acid transporter 1 (EAAT1) current after glutamate removal. (**A**) Anion currents in response to two pulses of rapid glutamate application (1 mM), with varying inter-pulse interval (pulse protocol shown at the top) under forward transport conditions. The intracellular solution contained 130 mM KSCN, the extracellular solution contained 140 mM sodium

*Figure 4 continued on next page*

*Figure 4 continued*

methanesulfonate (NaMes). (**B**) Similar experiment as in (**A**), but in the presence of 0.2 µM UCPH-101 (pre-incubated for 5 min, see open bar for timing of solution exchange, top of the figure). (**C**) Application of UCPH-101 alone did not induce any currents. (**D**) Time constants for the fast and slow phase of the current recovery, for the two exponential components. (**E–G**) Experiments similar to (**A–C**), but for the transport component of the current (the permeant intracellular anion, SCN⁻, was replaced with the non-permeant Mes⁻ anion). (**H**) Recovery of the transient current in the presence and absence of 0.2 µM UCPH-101. The membrane potential was 0 mV in all experiments.

The online version of this article includes the following source data and figure supplement(s) for figure 4:

**Source data 1.** Current traces for *Figure 3A,B,C,E,F and G*, and the original data for panels (D) and (H).

**Figure supplement 1.** UCPH-101 does not eliminate, but reduces capacitive charge movement caused by Na⁺ binding to the *apo*-form of excitatory amino acid transporter 1 (EAAT1).

**Figure supplement 1—source data 1.** Original current traces for *Figure 4—figure supplement 1A and B*, and the original data for panel (C).

---

assigned to steps associated with the glutamate translocation reaction (*Wang et al., 2018*). After pre-incubation and in the continuous presence of UCPH-101, both transient and steady-state current amplitude were reduced in a dose-dependent manner, as expected (*Figure 4B and F*, currents were fully eliminated at 40 µM UCPH-101), but at concentrations close to the apparent $K_i$, the decay of the current was also slowed (*Figure 4D*, $\tau$=11 ± 1.3, 16±2.1, 20±2.1 ms, at 0, 0.2, and 2 µM UCPH-101, respectively). The reduction of the decay rate of the transient current indicates a slowing of the translocation reaction in the presence of UCPH-101. In contrast to competitive inhibitors, UCPH-101 application in the absence of glutamate did not induce any currents, as expected for non-competitive inhibitors, which are not known to be able to block the glutamate transporter leak anion conductance (*Figure 4C and G*).

Upon removal of glutamate, recovery of the transient current upon glutamate application in a second pulse was relatively slow, with a time constant of 95±4 ms, most likely reflecting the turnover time of the transporter, which has to recycle its binding sites to outward-facing, after passing through a whole transport cycle. At 0.2 µM UCPH-101 recovery of the transient current after glutamate removal occurred with a time constant of 115±11 ms, as quantified in *Figure 4F and H*. Therefore, it appears that UCPH-101 does not significantly slow the recovery rate of the transient current. The most likely explanation is that, at low concentrations (<1 µM) UCPH-101 binding to EAAT1 and unbinding are very slow, with time constant of 10 s for binding, and 100 s for dissociation (*Abrahamsen et al., 2013*). Thus, within the time course of the paired-pulse experiment (1.4 s), UCPH-101-bound and -unbound transporter populations do no interconvert, and the time course of current recovery, thus, reflects that of the UCPH-101-free state.

Finally, we tested whether UCPH-101 binding affects how Na⁺ interacts with the *apo* (glutamate-free) form of the transporter. This question was raised from the crystal structures of EAAT1 in the presence of UCPH-101 (*Canul-Tec et al., 2017*). The structure shows Na⁺ bound only at the Na2 position (*Canul-Tec et al., 2017*). However, two more Na⁺ binding sites exist, Na1 and Na3 (*Canul-Tec et al., 2017*; *Reyes et al., 2009*; *Boudker et al., 2007*; *Yernool et al., 2004*; *Verdon and Boudker, 2012*; *Guskov et al., 2016*; *Wang et al., 2019*). As reported previously, Na⁺ binding to a glutamate-free form of EAAT2 and -3 is electrogenic (*Grewer et al., 2000*; *Watzke et al., 2001*; *Grewer et al., 2012*), generating transient currents due to Na⁺ movement into its binding site upon voltage jumps to more negative membrane potentials. Consistently, step changes of the membrane potential from –100 to +60 mV resulted in transient currents (*Figure 4—figure supplement 1*), which decayed with a time constant in the 0.5 ms range (140 mM Na⁺), in the absence of glutamate. Non-specific currents were subtracted by applying 200 µM TBOA. In the presence of UCPH-101, the charge movement was still present (*Figure 4—figure supplement 1B*), and the midpoint potential (a measure of Na⁺ affinity) was not significantly changed (*Figure 4—figure supplement 1C*). These results are consistent with the interpretation that Na⁺ binding (most likely to the Na1 and Na3 sites) is not inhibited by bound UCPH-101 and occurs with similar affinity as in the absence of the inhibitor.

## UCPH-101 is a partial inhibitor of ASCT2

In the next paragraphs, we describe electrophysiological measurements on ASCT2 in the presence of UCPH-101, to test whether the inhibition mechanism is conserved between the EAAT1 and ASCT2 members of the SLC1 family. To corroborate the residues directly contributing to the UCPH-101-bound state in EAAT1 suggested by docking, we used molecular dynamics (MD) simulations, to investigate the stability of UCPH-101 in its EAAT1 binding pocket (Materials and methods). From the EAAT1 structure bound to the substrate aspartate and UCPH-101 in the allosteric site (5LLM), we generated a model of EAAT1 inserted into a lipid bilayer in a water box. After equilibration, UCPH-101 remained stable in the original binding pocket in six 100 ns simulations (four representative ones shown in *Figure 5D*). We selected two conserved residues (Y127, M271; *Figure 5A* and *Figure 1—figure supplement 1*) to estimate distance changes to bound UCPH-101 together with RMSD calculations. Interestingly, the trajectories show that the position of UCPH-101 is stable relative to Y127 and M271 in 100 ns simulations, and these two residues stay in close contact with the UCPH-101 pyran ring oxygen and the amino group nitrogen (*Figure 5D*). It should be noted that UCPH-101 has two stereo-centers. For the simulations, we used the stereo-configuration analogous to the one from the EAAT1 UCPH-101-bound structure.

To investigate the stability of inhibitors in the predicted ASCT2 allosteric binding pocket, we also performed MD simulations on UCPH-101-bound ASCT2$_{WT}$ and ASCT2$_{F136Y/I237M}$. Residues F136Y and I237M (*Figure 5C*, *Figure 5—figure supplement 1*) were selected to evaluate distance changes to UCPH-101 during equilibration. The distance of UCPH-101 is relatively stable with respect to Y136 and M237 in 300–500 ns simulation trajectories (*Figure 5E*, *Figure 5—figure supplement 1*), while a slight, initial increase in the RMSD indicates relaxation of the structure compared to the initial, docked starting state. Two full dissociation events into the lipid bilayer were observed in six simulations within 300 ns for ASCT wild-type (*Figure 5E*), consistent with the predicted slow binding/dissociation kinetics, while no dissociation was observed in the double-mutant transporter (*Figure 5F*). The distance between selected atoms of UCPH-101 and ASCT2 were maintained within 6 Å in the double-mutant transporter, whereas in the wild-type these distances were 8–11 Å (*Figure 5E*), indicating that these interactions are less favorable in the wild-type transporter. Finally, as a control, we measured the distance between bound amino acid substrate and a residue in the binding site throughout the simulation. This distance remained stable throughout the simulations, until amino acid substrate dissociation occurred (dissociation is expected due to the low substrate affinity), illustrating convergence of our simulations and increasing the confidence in our approach.

To test the predictions from docking calculations and MD simulations, we next experimentally investigated the interaction of UCPH-101 with ASCT2$_{WT}$, and transporters with mutations to the analogous positions in ASCT2, using electrophysiological characterization of anion current, which is activated during substrate exchange. Anion conductance is an indirect measure of ASCT2 transport activity, but was previously shown to correlate with transport activity (direct amino acid transport measurements are shown below) (*Grewer and Grabsch, 2004*). Typical anion currents, using the highly permeant anion SCN$^-$, were inwardly directed, due to SCN$^-$ outflow through the ASCT2 anion conductance (*Figure 6A*). As expected, the anion currents increased with increasing serine concentration, with an apparent $K_m$ of 280±40 μM. When serine was co-applied together with UCPH-101, the current was inhibited, but not to 100%, even at saturating [UCPH-101] (*Figure 6B*). In addition, inhibition required much higher UCPH-101 concentrations than in EAAT1. The UCPH-101 dose-response relationship, at a constant serine concentration of 100 μM (*Figure 6C*) demonstrates maximum inhibition of about 45% at 200 μM UCPH-101, with an apparent $K_i$ value of 77±20 μM (*Figure 6C and D*). Although full saturation could not be reached due to UCPH-101 not being fully soluble at a concentration of 500 μM, the fit to the dose-response curve indicates partial inhibition even at saturating concentration (*Figure 6C*, red line). Maximum inhibition was not dependent on the serine concentration, as expected (*Figure 6D*). Furthermore, the apparent $K_i$ was only weakly dependent on [serine] (*Figure 7D*). From these results, we presume that UCPH-101 operates as a weak, non-competitive, partial inhibitor for the wild-type ASCT2 transporter.

As a direct measure of amino acid transport, we next tested the effect of UCPH-101 on amino acid (L-serine) uptake. The competitive inhibitor L-*cis*-BPE was used to determine the specific component of L-serine uptake (*Garibsingh et al., 2021*). We observed that serine uptake was also

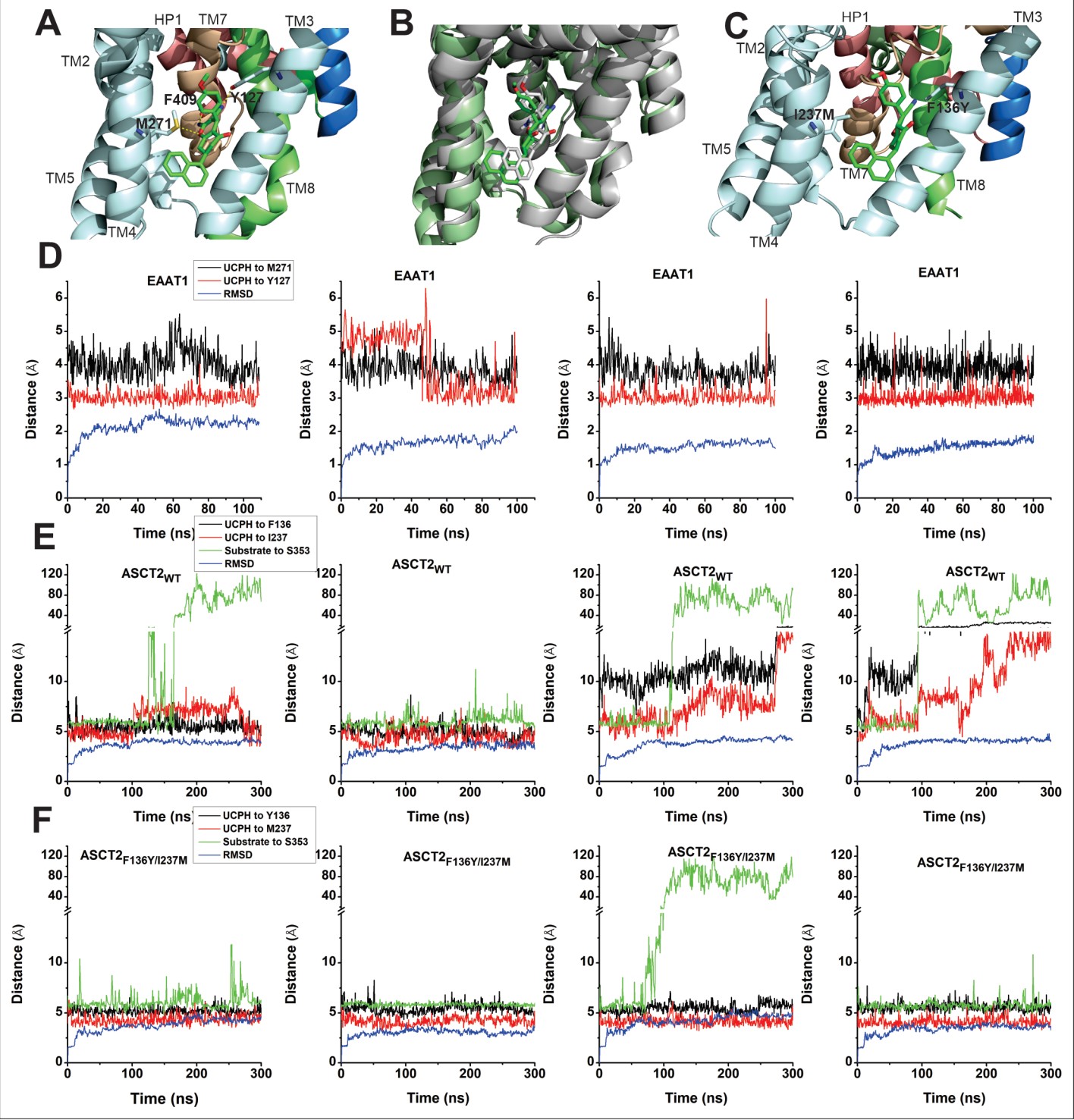

**Figure 5.** EAAT1, ASCT2$_{WT}$, and ASCT2$_{F136Y/I237M}$ residues contribute to UCPH-101 stability in the binding site. (**A**) Original state of the EAAT1-UCPH-101 complex from structure 5LLM (*Canul-Tec et al., 2017*). Y127 and M271 (EAAT1 sequence number) contribute to the EAAT1-UCPH-101 interaction. The binding state before (green) and after (gray) 100 ns molecular dynamics (MD) simulation are shown in (**B**). (**C**) Original state of the docked ASCT2$_{WT}$-UCPH-101 complex (7BCS) with modeled side chains. Trajectories for four independent simulations runs for EAAT1, ASCT2$_{WT}$, and ASCT2$_{F136Y}$/$_{I237M}$ (one UCPH-101 molecule at each subunit interface) were analyzed and are shown from (**D**) to (**F**) (distance calculation [red, black (UCPH-101), green (amino acid substrate)] and RMSD distance [blue]). For distance calculations, in EAAT1, we selected atoms Y127(CA) and M271(CA) for EAAT1 and (O) and (N) for UCPH-101 (Materials and methods) as reference atoms. In ASCT2$_{WT}$, we selected atoms F136(CZ) and I237(CD) for ASCT2$_{WT}$ and (N1) and (O2) for

*Figure 5 continued on next page*

*Figure 5 continued*

UCPH-101. In ASCT2$_{F136Y/I237M}$ distance calculation, we selected atoms Y136(OH) and M237(SD) from the transporter, and (N1) and (O2) from UCPH-101 as reference.

The online version of this article includes the following source data and figure supplement(s) for figure 5:

**Source data 1.** Trajectory and RMSD data for panels (D-F).

**Figure supplement 1.** Molecular dynamics (MD) simulations to 500 ns for ASCT2$_{WT}$ and ASCT2$_{F136Y/I237M}$.

**Figure supplement 1—source data 1.** Trajectory and RMSD data.

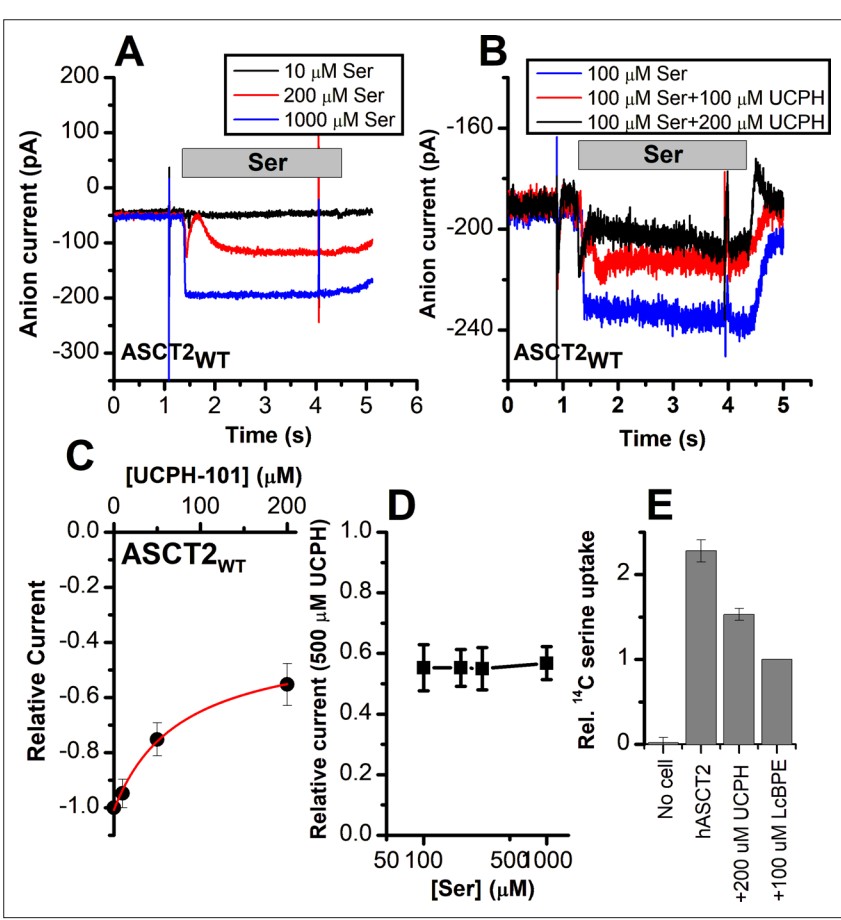

**Figure 6.** Alanine serine cysteine transporter 2 (ASCT2) amino acid-induced anion current is partially inhibited by UCPH-101. (**A**) Typical serine-induced ASCT2 anion current increases with increasing serine concentrations. The extracellular solution contained 140 mM sodium methanesulfonate (NaMes), with 130 mM NaSCN, and 10 mM Ser incorporated into the whole-cell recording electrode. The application time for serine is indicated by the gray bar. The apparent affinity for Ser were calculated as $K_m$ = 280 ± 40 µM. (**B**) Same experiment as in (**A**) at 100 µM Ser, but in the presence of UCPH-101 at three concentrations. (**C**) UCPH-101 dose-response curve at 100 µM Ser (*n*=15). (**D**) Relative current at saturating [UCPH-101], shown at varying Ser concentrations. (**E**) Uptake of $^{14}$C serine in hASCT2-expressing HEK293 cells in the absence and presence of 200 µM UCPH-101. Competitive ASCT2 inhibitor L-cis-BPE was used at a saturating concentration to determine specific uptake by ASCT2.

The online version of this article includes the following source data and figure supplement(s) for figure 6:

**Source data 1.** Original current traces for *Figure 6A and B*, and the original data for panels (C-E).

**Figure supplement 1.** ASCT2$_{F136Y/I237M}$ anion current is inhibited by UCPH-101 independent of voltage.

**Figure supplement 1—source data 1.** Current traces for *Figure 6—figure supplement 1A and B*, and the original data for panel (C).

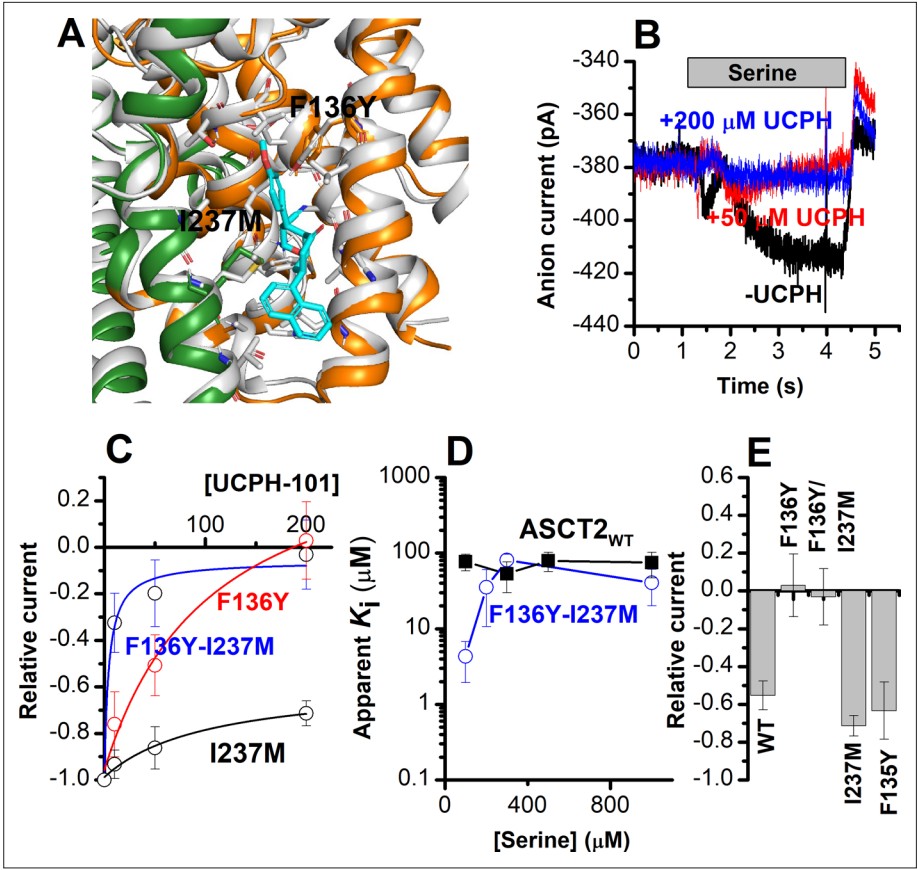

**Figure 7.** The F136Y/I237M ASCT2 double mutation restores complete inhibition of anion current by UCPH-101. (**A**) Predicted binding pose of UCPH-101 (cyan sticks) to the F136Y/I237M-double-mutant ASCT2 transporter (the trimerization and transport domains are highlighted in dark green and dark orange, respectively) in overlay with EAAT1 (white). (**B**) Typical whole-cell current recordings for ASCT2$_{F136Y/I237M}$ at 100 μM serine. (**C**) Dose-response curves for ASCT2$_{WT}$ (black), and the F136 (red) and F136Y/I237M (blue) mutant transporters. Blocking effects were measured at a serine concentration of 100 μM and at 0 mV transmembrane potential. (**D**) UCPH-101 apparent inhibition constant ($K_i$) plotted as a function of the serine concentration. (**E**) Comparison of steady-state current at 100 μM Ser and 200 μM UCPH-101, wild-type ASCT2 currents without UCPH-101 were set as reference to 1, error bars represent ± SD.

The online version of this article includes the following source data for figure 7:

**Source data 1.** Current traces for **Figure 7B**, and the original data for panels (C-E).

inhibited by UCPH-101 at a saturating concentration (**Figure 6E**); however, similar to the electrophysiological data, this inhibition was not complete, with 58% of the specific uptake being blocked by UCPH-101.

To experimentally test the hypothesis, predicted by docking (see above), that residues F136 and I237 contribute to UCPH-101 binding, four mutant ASCT2s were generated based on EAAT1, F135Y, F136Y, I237M, and the F136Y/I237M double mutant (predicted docking pose for the double-mutant ASCT2 is shown in **Figure 7A**). Before testing with UCPH-101, the mutant transporters were functionally analyzed by whole-cell current recording experiments, to check whether the mutations altered transporter functionality. All four mutant transporters exhibited wild-type-like anion currents upon substrate application. Mutant transporters with F136Y and F136Y/I237M substitutions changed the substrate selectivity, that is, in contrast to ASCT2$_{WT}$, anion current could not be observed after application of glutamine (**Table 1**). In addition, the F136Y mutation decreased the serine apparent affinity from a $K_m$ of 280±40 μM (ASCT2$_{WT}$) to 790±130 μM (**Table 1**). None of the other mutant transporters showed a significant effect on the $K_m$ for Ser/Ala current activation. It remains unclear why mutations to these sites have some effect on substrate selectivity/affinity, despite being far removed from the substrate binding site. We hypothesize that indirect effects of the mutations are involved.

**Table 1.** Substrate selectivity of all alanine serine cysteine transporter 2 (ASCT2) mutants.

| Substrate binding affinity (µM) | Ala | Ser | Gln |
|---|---|---|---|
| WT | 290±50 | 280±40 | 190±30 |
| F135Y | 300±60 | 180±10 | 190±50 |
| F136Y | 280±70 | 790±130 | Insensitive |
| I237M | 230±20 | 300±70 | 270±50 |
| F136Y/I237M | 300±130 | 310±100 | Insensitive |

The online version of this article includes the following source data for table 1:

**Source data 1.** Original data for all Km values listed in *Table 1*.

We next determined the effect of UCPH-101 on the mutant ASCT2s. Serine-induced anion currents were inhibited by UCPH-101 in all four mutant transporters (*Figure 7B–E*), however, to varying degrees. Original UCPH-101 inhibition data for the double-mutant transporter and dose-response curves are shown in *Figure 7B and C*. In contrast to the wild-type transporter, UCPH-101 was able to block virtually 100% of the serine-induced inward current at a concentration of 200 µM in both F136Y and the double-mutant transporter (*Figure 7B and C*). However, no complete block was observed in ASCT2$_{I237M}$, and the UCPH-101 interaction was of low affinity. (*Figure 7C and D*). For the double-mutant F136Y/I237M transporter, the apparent UCPH-101 affinity was significantly increased over the wild-type transporter, at least at low [serine]. As shown in *Figure 7B and D*, in ASCT2$_{F136Y/I237M}$, UCPH-101 fully blocked the anion currents above 100 µM UCPH, resulting in an apparent $K_i$ value of 4.3±2.4 µM (*Figure 7C and E*).

Subsequently, we compared the UCPH-101 apparent $K_i$ as a function of the substrate concentration (*Figure 7D*). For the double-mutant transporter, the apparent $K_i$ increased with increasing [serine] at low concentration but leveled off at high concentrations, which is unexpected for a purely noncompetitive inhibition, but consistent with a mixed inhibition mechanism, in which the inhibitor binds somewhat more strongly to the *apo*-transporter than the substrate-bound form.

Finally, we tested the UCPH-101 inhibition of the double-mutant transporter at varying membrane potentials. Serine-induced anion currents were significantly smaller than in the presence of UCPH-101 (*Figure 6—figure supplement 1B*), compared to control (*Figure 6—figure supplement 1A*). Inhibition was more pronounced at negative membrane potentials, as shown in *Figure 6—figure supplement 1C*.

## UCPH-101 slows onset and recovery of anion currents induced by substrate in wild-type and ASCT2$_{F136Y/I237M}$

When serine was applied rapidly to ASCT2$_{WT}$ and ASCT2$_{F136Y/I237M}$, in the presence of intracellular SCN$^-$ (substrate-induced anion current), the current was activated immediately (*Figures 8A and 9A*), within the time resolution of the solution exchange (5–10 ms). In some experiments, a small, but significant transient current component was observed (*Figure 9A*), due to the population of an intermediate along the translocation pathway with high anion conductance, as proposed previously (*Zander et al., 2013*). Unlike EAAT1 in forward transport mode, ASCT2 anion current also deactivated rapidly upon serine removal, and recovered very quickly when a second pulse of serine was applied (*Figure 8A*, time of decay and recovery all within the time resolution of solution exchange). Similar results were observed for ASCT2$_{F136Y/I237M}$ (*Figure 9A*). However, when serine was applied in the presence of 100 µM UCPH-101 (pre-incubated for 5 min), steady-state anion current was inhibited, as expected, but in addition a significant slowing of anion current rise after serine application, decay after serine removal, and recovery after a second pulse of serine application was observed, for both ASCT2$_{WT}$ (*Figure 8B*) and ASCT2$_{F136Y/I237M}$ (*Figure 9B*). The time constant of the responses, which were biphasic and required fitting with two exponential components, are summarized in *Figures 8D and 9D*, demonstrating a significant increase in the time constant for recovery, in particular for the fast component of the recovery. These results suggest that UCPH-101 not only inhibits current amplitude, but also slows current kinetics and serine exchange turnover (as measured by current recovery), as expected for a fast-binding inhibitor that shows rapid pre-equilibrium between the UCPH-101-bound and -unbound transporter states, with respect to the substrate-induced transporter kinetics. Using pre-equilibrium conditions, the following equation can be used to describe the kinetics of the translocation equilibrium:

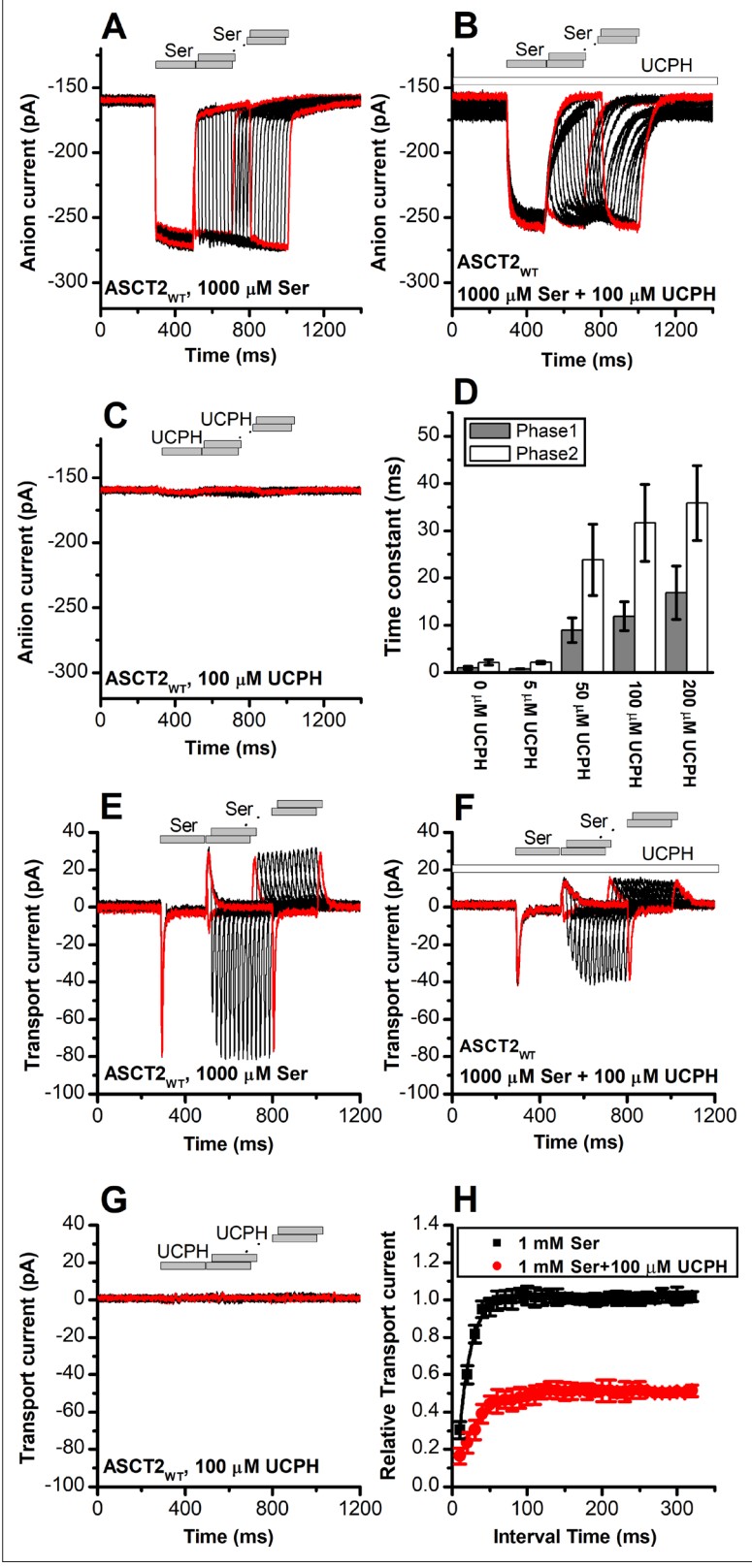

**Figure 8.** UCPH-101 slows kinetics of alanine serine cysteine transporter 2 (ASCT2) current onset and recovery after amino acid removal. (**A**) Anion currents in response to two pulses of rapid serine application (1 mM), with varying inter-pulse interval (pulse protocol shown at the top) under homo-exchange conditions. The intracellular solution contained 130 mM NaSCN/10 mM serine, the extracellular solution contained 140 mM sodium methanesulfonate

*Figure 8 continued on next page*

*Figure 8 continued*

(NaMes). (**B**) Similar experiment as in (**A**), but in the presence of 100 µM UCPH-101 (pre-incubated for 5 min, see open bar for timing of solution exchange, top of the figure). (**C**) Application of UCPH-101 alone did not induce any currents. (**D**) Time constants for the fast and slow phase of the current recovery, for the two exponential components. (**E–G**) Experiments similar to (**A–C**), but for the transport component of the current (the permeant intracellular anion, SCN⁻, was replaced with the non-permeant Mes⁻ anion). (**H**) Recovery of the transient current in the presence and absence of 100 µM UCPH-101. The membrane potential was 0 mV in all experiments.

The online version of this article includes the following source data for figure 8:

**Source data 1.** Original current traces for *Figure 8A,B,C,E,F and G*, and the original data for panels (D) and (H).

$$k_{\mathrm{obs}} = k_{\mathrm{f}} \frac{[S]_{\mathrm{o}}}{\left([S]_{\mathrm{o}} + K_{\mathrm{m}}\right)} \frac{K_{\mathrm{i}}}{\left([I] + K_{\mathrm{i}}\right)} + k_{\mathrm{r}} \frac{[S]_{\mathrm{i}}}{\left([S]_{\mathrm{i}} + K_{\mathrm{m}}\right)} \frac{K_{\mathrm{i}}}{\left([I] + K_{\mathrm{i}}\right)} \tag{2}$$

Here, $k_{\mathrm{obs}}$ is the observed rate constant for the rise or decay of the anion current (i.e. equilibration of the translocation equilibrium), which is a sum of the rate constants for forward ($k_f$) and reverse ($k_r$) translocation. $[S]_{\mathrm{o}}$ and $[S]_{\mathrm{i}}$ are the external and internal substrate concentrations, respectively, $[I]$ is the UCPH-101 concentration and $K_{\mathrm{m}}$ and $K_{\mathrm{i}}$ have their regular meaning. When amino acid substrate is rapidly removed, $[S]_{\mathrm{o}}$ becomes 0 and it is possible to isolate the reverse translocation rate constant.

## Rapid charge movements caused by ASCT2 substrate translocation are inhibited by UCPH-101

Similar to EAAT1, rapid external serine application to ASCT2 elicits transient inward transport currents (in the absence of permeable anion), which decay within 20 ms back to baseline (*Figure 8E*) in the absence of UCPH-101. In contrast to EAAT1, ASCT2 is an amino acid exchanger, therefore, the experiments were performed in the exchange mode (saturating [Na⁺] and [amino acid] on the intracellular side of the membrane). Because ASCT2 amino acid exchange is electroneutral at equilibrium (equal net rate of forward and reverse charge movement), no steady-state current was observed under these conditions (*Figure 8E*). However, upon rapid removal of extracellular amino acid, transient outward current was observed (*Figure 8E*), due to the re-equilibration of the homo-exchange translocation equilibrium to favor the outward-facing conformation in the absence of external amino acid. In contrast to the EAATs, Na⁺ apparent binding affinity to the *apo*-form of the ASCT2 transporter is high ($K_{\mathrm{m}} < 1$ mM [*Zander et al., 2013*]). Therefore, the Na⁺ binding equilibria of the amino acid-free transporter are fully saturated at 140 mM external [Na⁺], and the amino acid-induced charge movement can only be caused by Na⁺ binding to the amino acid-bound transporter, substrate translocation, or, most likely, both of those processes. Alanine and serine exchange was previously reported to be rapid in ASCT2 (*Wang et al., 2022*). Consistent with this proposal, transient transport current recovered much faster than in EAAT1 after amino acid removal (*Figure 8E and H*), with a time constant of 15±0.2 ms (*Figure 8H*).

When the same paired-pulse serine solution exchange protocol was applied to ASCT2 in the presence of UCPH-101, similar transient currents were observed (inward upon serine application and outward upon removal), but the amplitude of the transient currents was significantly reduced (*Figure 8F*). However, similar to the observations from the anion currents, UCPH-101 was unable to completely block transient transport current. Residual inward transient current amplitude at 100 µM [UCPH-101] was 50% compared to control. As expected, UCPH-101 application alone did not generate any transient currents (*Figure 8G*). Recovery time of the current was increased twofold to 30±0.4 ms at 100 µM UCPH-101 (*Figure 8H*). Overall, these data indicate that UCPH-101 reduces, but does not eliminate charge movements induced by Na⁺ binding to the amino acid-bound transporter, and/or substrate translocation across the membrane, in other words, the steps needed for amino acid transport by ASCT2.

Similar results were obtained for ASCT2$_{\mathrm{F136Y/I237M}}$ (*Figure 9*). As in the case for anion current, UCPH-101 also significantly slowed the kinetics of decay and recovery of the transient transport current (*Figure 9E–H*), indicating that UCPH-101, due to decreased affinity compared to EAAT1, also acts like a rapidly equilibrating inhibitor in the double-mutant ASCT2 transporter.

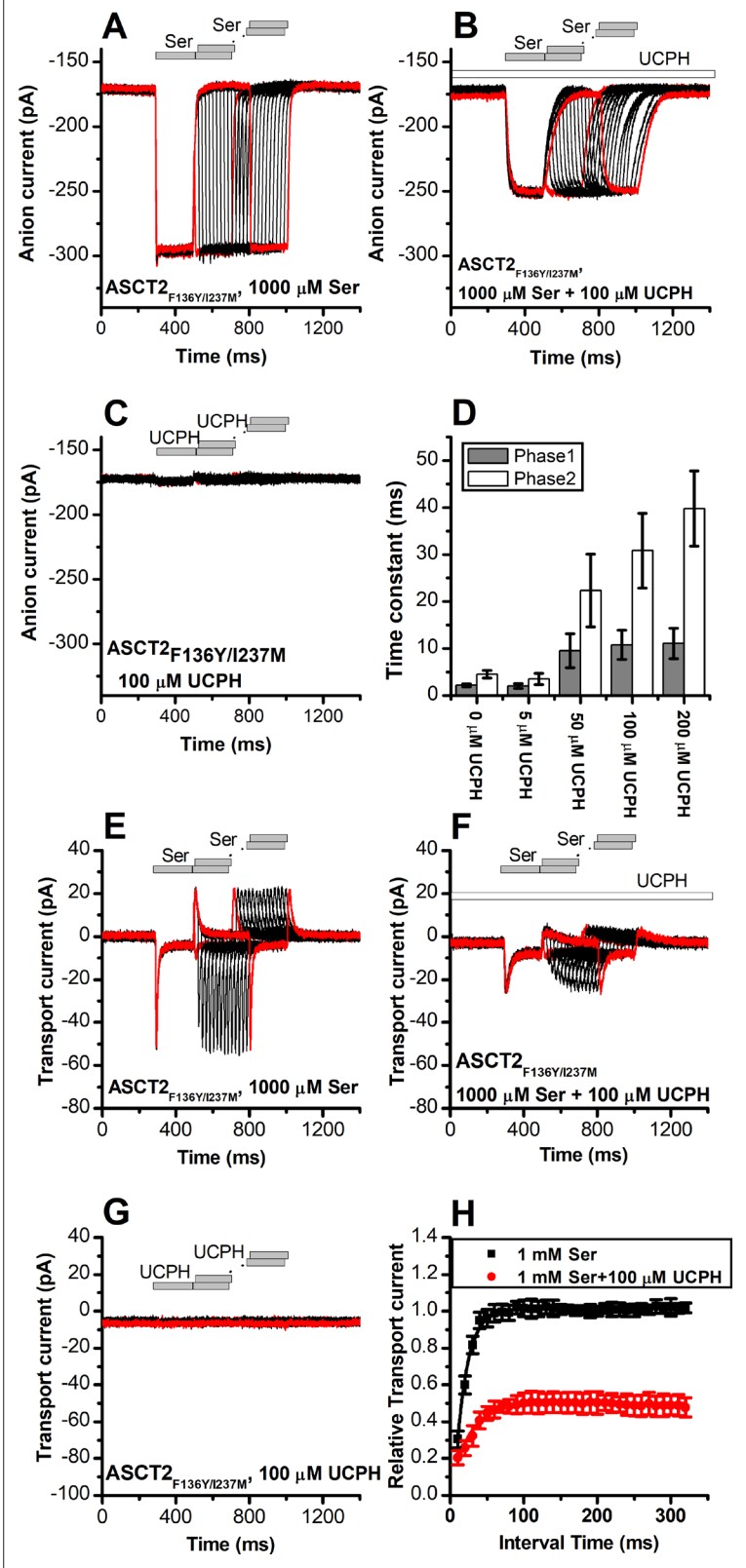

**Figure 9.** UCPH-101 slows kinetics of ASCT2$_{F136Y/I237M}$ current onset and recovery after amino acid removal. (**A**) Anion currents in response to two pulses of rapid serine application (1 mM), with varying inter-pulse interval (pulse protocol shown at the top) under homo-exchange conditions in an ASCT2$_{F136Y/I237M}$-expressing cell. The intracellular solution contained 130 mM NaSCN/10 mM serine, the extracellular solution contained 140 mM

*Figure 9 continued on next page*

*Figure 9 continued*

sodium methanesulfonate (NaMes). (**B**) Similar experiment as in (**A**), but in the presence of 100 µM UCPH-101 (pre-incubated for 5 min, see open bar for timing of solution exchange, top of the figure). (**C**) Application of UCPH-101 alone did not induce any currents. (**D**) Time constants for the fast and slow phase of the current recovery, for the two exponential components. (**E–G**) Experiments similar to (**A–C**), but for the transport component of the current (the permeant intracellular anion, SCN⁻, was replaced with the non-permeant Mes⁻ anion). (**H**) Recovery of the transient current in the presence and absence of 100 µM UCPH-101. The membrane potential was 0 mV in all experiments.

The online version of this article includes the following source data and figure supplement(s) for figure 9:

**Source data 1.** Original current traces for *Figure 9A,B,C,E,F and G*, and the original data for panels (D) and (H).

**Figure supplement 1.** UCPH-101 does not eliminate, but reduces capacitive charge movement caused by Na⁺ binding to the *apo*-form of $ASCT2_{F136Y}$ and $ASCT2_{F136Y/I237M}$.

**Figure supplement 1—source data 1.** Original current traces for *Figure 9—figure supplement 1A and B*, and the original data for panels (C) and (D).

## UCPH-101 effect on Na⁺ binding to the *apo*-form of $ASCT2_{F136Y/I237M}$

It was previously shown that voltage jumps to substrate-free ASCT2 result in transient charge movement from Na⁺ binding/dissociation (*Zander et al., 2013*) similar to EAATs (*Grewer et al., 2012*). When a step change in the membrane potential was applied to $ASCT2_{F136Y/I237M}$-expressing cells (*Figure 9—figure supplement 1A*), transient currents were observed. In these recordings, non-specific currents were subtracted by γ-(4-biphenylmethyl)-L-proline, which is a commercially available, competitive inhibitor of ASCT2 (*Singh et al., 2017*). In the presence of UCPH-101, the transient currents were reduced at all membrane potentials (*Figure 9—figure supplement 1B*). Q-V relationships are shown in *Figure 9—figure supplement 1C*, suggesting that Na⁺ binding and release steps in *apo*-transporter were inhibited by UCPH-101, but not eliminated, similar to our results obtained with EAAT1 shown above (*Figure 4—figure supplement 1*). Since under substrate-free conditions the translocation is not going to proceed, these results suggest that UCPH-101 likely affects Na⁺ binding to the proposed Na3/Na1 binding sites.

## Identification of a non-UCPH-101-like allosteric inhibitor of ASCT2

Our next goal was to confirm the location of the allosteric binding site by identifying compounds chemically different from UCPH-101 that interact with the ASCT2 allosteric binding site. We therefore employed virtual screening of the ZINC20 lead-like library (3.8 million compounds) to identify potential binders (Materials and methods). We analyzed the top scoring compounds for closer inspection and selected compounds that made predicted hydrogen bonding interactions to side chains of the allosteric binding site. From the best compounds found in the virtual screening, 11 were tested experimentally. We tested the inhibitory effect at 100 µM of serine. One of these compounds (#302, *Figure 10A*, docking pose shown in *Figure 10B*) inhibited ASCT2 anion current (*Figure 10G and H*), with an apparent $K_i$ of 238±58 µM, as well as transient transport current elicited by the transported substrate L-serine. In addition, 300 µM compound #302 inhibited the peak amplitude of ASCT2 transient transport current after rapid application of 1 mM serine to 48 ± 6% of the control current (*Figure 10C and D*). In contrast to UCPH-101, however, the presence of compound #302 did not have significant effect on the decay kinetics of the transient current, or the recovery of transient transport current after removal of amino acid substrate ($\tau$=19 ± 2.0 ms in the absence and 39±1.4 ms in the presence of #302, recovery kinetics shown in *Figure 10F*). As for UCPH-101, application of compound #302 alone (in the absence of serine) did not induce any current response (*Figure 10E*). To test whether compound #302 binds to the predicted allosteric ASCT2 binding site, we determined the apparent $K_i$ with $ASCT2_{F136Y/I237M}$ (*Figure 10G*) as 90±30 µM, demonstrating that the double mutation in the binding site increases the apparent affinity for compound #302, as was expected from the UCPH-101 results. Furthermore, the inhibition mechanism of compound #302 was found to be non-competitive (*Figure 10I*). Finally, compound #302 is able to inhibit ASCT1 substrate-induced anion current. This result was expected, since the allosteric binding site is largely conserved between ASCT1 and -2 (I237 is substituted by L in ASCT1, *Figure 1—figure supplement 1*) Notably, while compound #302 is not an optimized allosteric inhibitor of ASCT2, it provides a further validation for the location

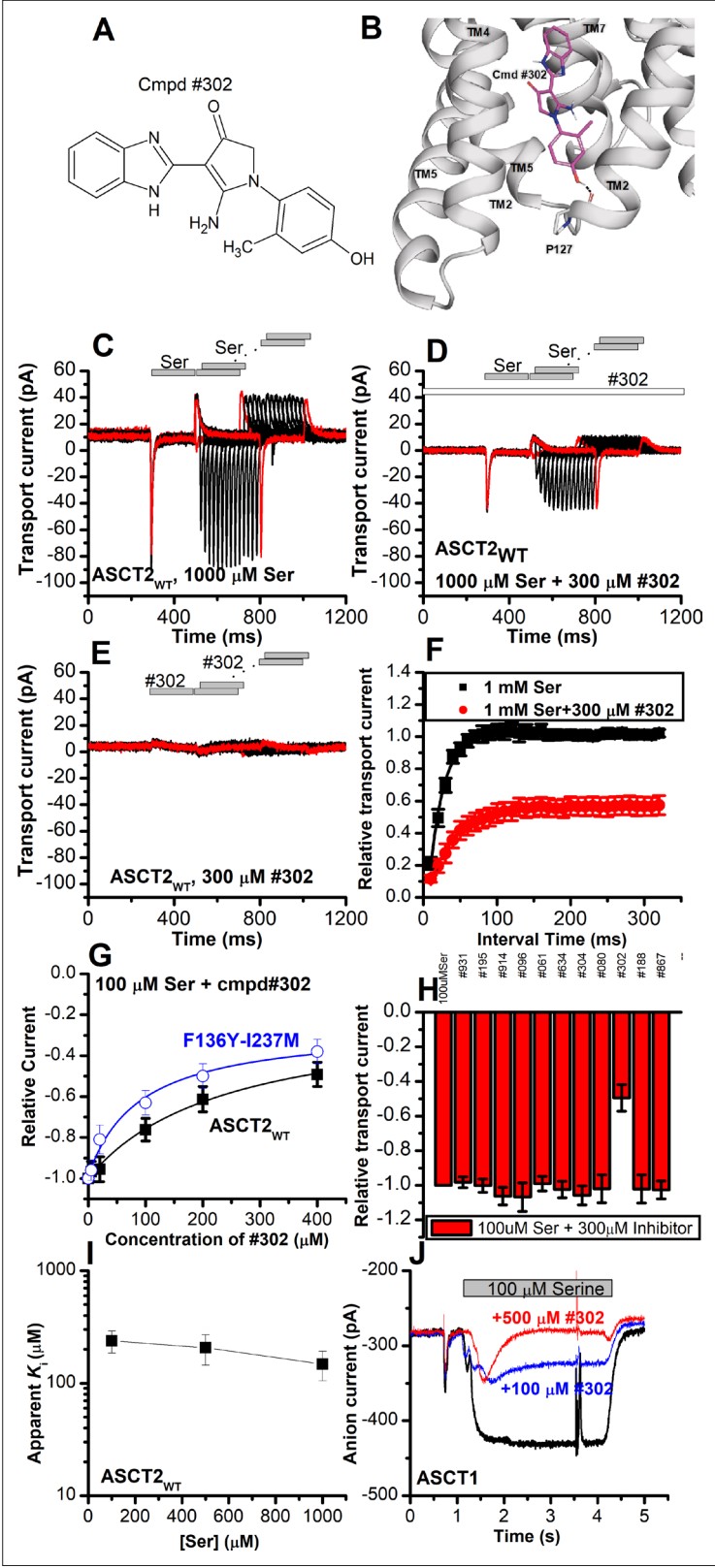

**Figure 10.** Novel Cmpd #302, identified by docking analysis, is a partial inhibitor of alanine serine cysteine transporter 2 (ASCT2) transient transport current. (**A**) Chemical structure of compound #302. (**B**) Predicted docking pose of compound #302 in the ASCT2 allosteric binding site. (**C**) Anion currents in response to two pulses of rapid serine application (1 mM), with varying inter-pulse interval (pulse protocol shown at the top) under homo-exchange

*Figure 10 continued on next page*

*Figure 10 continued*

conditions in an ASCT2-expressing cell. The intracellular solution contained 130 mM sodium methanesulfonate (NaMes)/10 mM serine, the extracellular solution contained 140 mM NaMes. (**D**) Similar experiment as in (**A**), but in the presence of 300 µM #302 (pre-incubated for 5 min, see open bar for timing of solution exchange, top of the figure). (**E**) Application of #302 alone did not induce any currents in ASCT2. (**F**) Recovery of the transient current in the presence and absence of 300 µM #302. The membrane potential was 0 mV in all experiments. (**G**) UCPH-101 dose-response curve at 100 µM Ser for ASCT2$_{WT}$ (black, *n*=18) and the F136Y-I237M double-mutant transporter (blue, *n*=14). (**H**) Comparison of steady-state current at 100 µM Ser and 300 µM inhibitors, wild-type ASCT2 currents without inhibitor were set as reference to 1, error bars represent ± SD. (**I**) Apparent $K_i$ of ASCT2$_{WT}$ for #302 as a function of the serine concentration, indicating a non-competitive inhibition mechanism. (**J**) Inhibition of ASCT1 serine-induced anion current (100 µM serine, time of application indicated by the gray bar). The voltage was 0 mV. Serine and #302 were applied at the same time.

The online version of this article includes the following source data for figure 10:

**Source data 1.** Original current traces for *Figure 10C,D,E and J*, and the original data for panels (F-I).

---

of the binding site as well as a starting point for developing future tool compounds targeting ASCT2 allosterically.

## Discussion

The work presented here allows us to draw four major conclusions about the mechanism of interaction of UCPH-101, and possibly other allosteric modulators, with the SLC1 family of transporters. First, UCPH-101 inhibits substrate transport by interfering with conformational changes associated with the substrate translocation reaction, rather than substrate and Na$^+$ binding. Second, the general location of the allosteric binding site of UCPH-101 is conserved between the SLC1 family members EAAT1 (glutamate transporter) and ASCT2 (neutral amino acid transporter), and possibly ASCT1, while the transporter side chains contributing to the binding pocket are not fully conserved. Third, occupancy of the allosteric binding site does not necessarily result in full inhibition of substrate transport, it can also lead to a partially active state, as seen for ASCT2$_{WT}$'s interaction with UCPH-101. Our results are therefore in agreement with previous mutagenesis studies (*Abrahamsen et al., 2013*), suggesting that allosteric compounds may be found that modulate transport activity, rather than completely blocking it, even raising the possibility of allosteric modulators that increase transport activity by increasing the substrate translocation rate. Fourth, the allosteric ASCT2 binding site can be targeted by compounds that are non-UCPH-101-like in structure. Overall, these are novel findings that would warrant the further investigation of the allosteric binding site and the development of future, more potent compounds targeting it.

### UCPH-101 interaction with EAAT1

This report provides additional mechanistic information on the action of the allosteric inhibitor, UCPH-101 on EAATs. In agreement with previous studies, UCPH-101 shows the hallmark of a non-competitive inhibitor of EAAT1, that is, a lack of effect of substrate concentration on the inhibitor $K_i$. In addition, Na$^+$ affinity of the *apo*-transporter seems to be unaffected. We also concluded that UCPH-101 does not affect the substrate binding steps, mainly by analysis of the kinetics of pre-steady-state currents in the presence of UCPH-101. These conclusions are in agreement with the EAAT1 structure in complex with both UCPH-101 and a competitive inhibitor, TFB-TBOA (*Shimamoto et al., 2004*), which demonstrated that the allosteric and substrate binding sites can be occupied at the same time.

In contrast to substrate binding, we propose that the substrate translocation step is the major partial reaction that is blocked in the UCPH-101-bound state. This interpretation is based on the significant reduction of the amplitude and the rate of the transient current decay in response to glutamate application (*Figure 4*), which for EAAC1 was previously assigned to the glutamate translocation step. From a structural viewpoint, this interpretation is logical, and was proposed by Canul-Tec et al. in 2017 from the EAAT1 structure in complex with UCPH-101 (*Canul-Tec et al., 2017*). Since binding occurs at the interface between transport and the trimerization domain, it is likely that transport domain movements relative to the trimerization domain are affected by the occupied allosteric site, either fully inhibiting those movements or slowing them to attain partial activity.

The effect of UCPH-101 on the glutamate-induced anion current of EAAT1 fits with this interpretation. It was shown previously that UCPH-101 exhibits very slow binding and dissociation kinetics (on the 10 s second time scale), much slower than the partial reactions of the glutamate-induced transport process and the turnover rate (in the 10–30 s$^{-1}$ range). Therefore, it is likely that the UCPH-101-free and -bound states interconvert slowly on the time scale of transport. Thus, UCPH-101 reduces the amplitude of glutamate-induced anion and transport current, without a significant effect on the observed kinetics associated with these reactions.

## Conserved modulatory allosteric mechanism in ASCT2

The development of novel inhibitors are essential steps for potential treatment of diseases, especially for ASCT2 transporter. In previous reports, overexpression of ASCT2 was observed in several cancer cell lines (*Bröer et al., 1999*; *Son et al., 2013*; *Wahi and Holst, 2019*; *Kanai et al., 2013*). Among the several published ASCT2 inhibitors, most of them target the substrate binding site. However, allosteric modulators would be useful because of the independence of their inhibitory effect on substrate concentration. The known allosteric binding pocket of EAAT1 as well as the biomedical importance of ASCT2 prompted us to investigate whether a similar allosteric mechanism is present in ASCT2; and, indeed, UCPH-101 was found to interact with this binding site. Therefore, the importance of this report lies in the generation of a new structural model proposed here to develop future novel allosteric modulators with the potential for higher affinity and specificity for ASCT2 in future compounds.

In the EAAT members of the SLC1 family, UCPH-101 was found to be specific for EAAT1. For example, in wild-type Glt-1 (EAAT2), no inhibition was observed, unless EAAT1/EAAT2 chimeras were generated that contained the EAAT1 N-terminal six transmembrane domains (*Abrahamsen et al., 2013*). In contrast to the EAATs, however, it was not known whether UCPH-101 can interact with the ASCT members of the SLC1 family. Our results reported here are, to our knowledge, the first to show that UCPH-101 inhibits ASCT2, providing evidence for the existence of the allosteric binding site in the same location as in EAAT1. While UCPH-101 is only a partial inhibitor of wild-type ASCT2, and its apparent inhibition constant is relatively high (low affinity, in the range of 100 µM), it demonstrates the principle of allosteric inhibition, or modulation of ASCT function through the allosteric binding site. Importantly, we were able to restore complete inhibition of substrate-induced anion current in mutant ASCT2 transporters, in which the proposed UCPH-101 binding site was modified to resemble that of EAAT1. In the double-mutant ASCT2$_{F136Y/I237M}$ the affinity was also increased compared to the wild-type transporter, although not reaching the high-affinity levels seen in EAAT1.

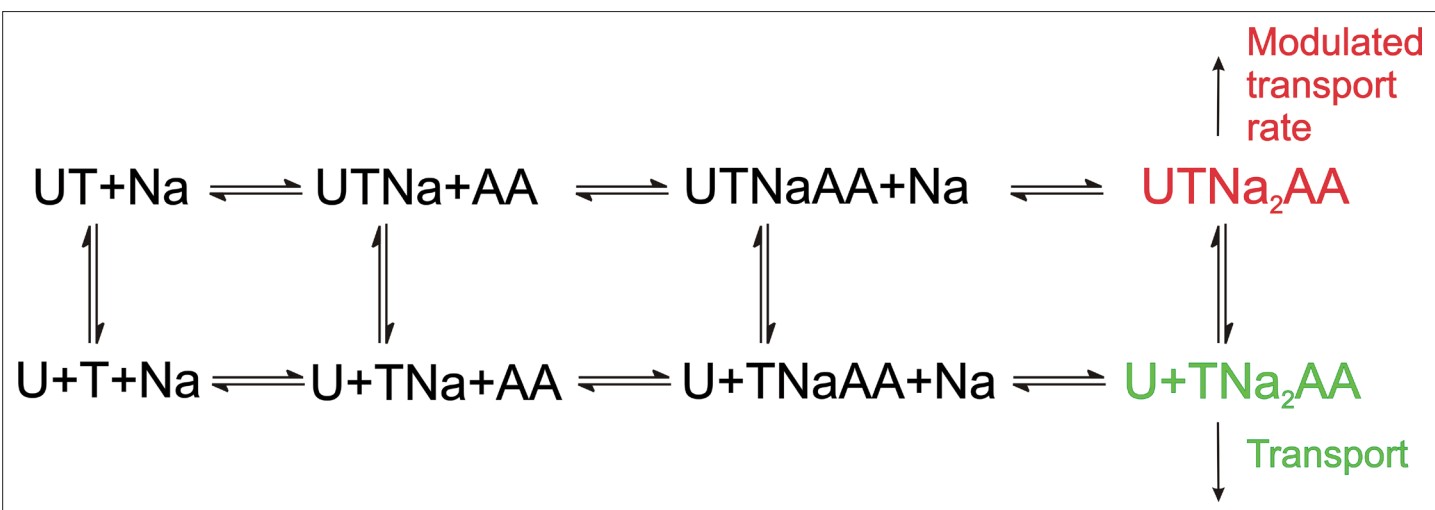

**Figure 11.** Proposed simplified mechanism of inhibitors (**U**) targeting the allosteric binding site interacting with alanine serine cysteine transporter 2 (ASCT2) and excitatory amino acid transporter 1 (EAAT1) transporters (**T**). The amino acid substrate is abbreviated as AA. Potential charges of the cation (Na$^+$) and the transporter/substrate were omitted for simplicity, and only two of the three Na$^+$ binding steps are shown. For UCPH-101 interaction with EAAT1 the UCPH binding/dissociation process (transition from the first to the second row of the scheme) is slow, while this transition is fast for ASCT2, accounting for the differential effect of UCPH-101 on transport kinetics of both transporters.

As expected, UCPH-101 shows the hallmarks of a non-competitive inhibitor for ASCT2$_{WT}$, however, for the double-mutant transporter a mixed inhibition mechanism was observed, pointing to higher affinity of the *apo*-transporter for UCPH-101, compared to the substrate-bound form. The reason for this is not known, but somehow the UCPH-101 binding site in this mutant transporter must communicate with the substrate binding site. A proposed kinetic mechanism for inhibition of EAAT1/ASCT2 by UCPH-101 is shown in *Figure 11*, in analogy to previously published alternating access mechanisms for the SLC1 family (*Kanner and Bendahan, 1982*; *Kanner and Zomot, 2008*). Here, UCPH-101 can bind to all of the states the transporter can access during the transport cycle, including substrate- and Na$^+$-bound states. The reaction that is affected by occupation of the allosteric site is the translocation reaction, or reactions closely associated with it, such as Na$^+$ binding to the substrate-bound transporter. In EAAT1, due to the high affinity of the compound, equilibration of the UCPH-101-bound and -unbound states is slow, thus, UCPH-101 inhibits the transient and steady-state currents, with little effect on apparent transport kinetics. In essence, UCPH-101 acts by removing active EAAT1 transporters from the pool, thus inhibiting the overall transport rate. On the other hand, equilibration of the UCPH-101-bound and -unbound states is fast in ASCT2, therefore, kinetics of substrate-induced current activation and recovery are slowed in the presence of inhibitor. Thus, the kinetic mechanism shown in *Figure 11* can account for the subtleties of kinetic differences in UPCPH-101 effects between EAAT1 and ASCT2.

One limitation with current compounds targeting ASCT2 for cancer therapy is that they are amino acid-like compounds targeting the substrate binding site, and thus have poor pharmacokinetics. Interestingly, we were able to identify a compound with a structure unrelated to UCPH-101, compound #302, which inhibited wild-type ASCT2 in a non-competitive manner, although at relatively high concentrations and, like UCPH-101, showing only partial inhibition of the transient transport current elicited by substrate application. This compound was found from virtual screening against the putative allosteric binding site. Interestingly compound #302 also inhibited ASCT1, in which the critical residues in the allosteric binding site are similar to ASCT2. Like UCPH-101, compound #302 is relatively hydrophobic and harbors an amino function. However, it also has a OH group, suggesting that additional hydrophilic interaction may be important in the allosteric binding site. These results further support the presence of the allosteric binding site in ASCT2 and suggest that it may be possible to identify allosteric inhibitors of ASCT2 with novel scaffolds, opening new avenues for future structure-activity relationship studies, to improve affinity and specificity.

ASCT2 has recently received attention due to its involvement in neutral amino acid homeostasis in cells, in particular cancer cells. It was shown that down-regulation of ASCT2 function either by siRNA or by competitive inhibitors, can block the growth of cancer cells in cell-line models, as well as in vivo tumor transplants (*Ni et al., 2019*; *Wang et al., 2015*). Competitive inhibitors have the disadvantage of becoming less effective as the amino acid concentration rises. Having allosteric inhibitors available would result in amino acid concentration independent block of ASCT2 function, which would be much more desirable for pharmacological applications. The existence of the allosteric binding site on ASCT2 raises the possibility that such compounds could be developed, which, in contrast to UCPH-101, would result in complete inhibition of transporter function.

## Conclusions

The results presented in this report reveal a conserved allosteric inhibition mechanism for SLC1 members for the first time. The results suggest that the major effect of the compound, once bound to the binding site at the interface between the transport and trimerization domain, is to block reactions associated with substrate translocation and anion conductance activation. In contrast, other partial reactions, such as substrate and Na$^+$ binding can still take place, although the electrogenicity of Na$^+$ binding is reduced, pointing to a slight conformational change of the substrate-free transporter state induced by UCPH-101. While UCPH-101 is a partial and weak inhibitor of wild-type ASCT2, complete inhibition could be restored by mutating the UCPH-101 binding site to one that resembles EAAT1. Therefore, our results suggest that ASCT2 also harbors the allosteric binding site, however, with differences in the amino acid side chains contributing to the site. These results raise the possibility that this allosteric binding site can be exploited by future compounds specifically targeting ASCT2. Our results involving the novel inhibitor compound #302 further confirm the location of the allosteric binding site in ASCT and suggest that compounds with varying chemical scaffolds could be found that modulate

or completely inhibit SLC1 activity. In light of the emerging role of ASCT2 as an anti-cancer drug target, the future characterization of this allosteric site appears promising.

## Materials and methods

### Cell culture and transfection

HEK293 cells (American Type Culture Collection, ATCC No. CRL 1573) were cultured as described previously (*Wang et al., 2022*). The cell line was authenticated by STR profiling (ATCC). Cells were tested for mycoplasm contamination using 4',6-diamidino-2-phenylindole staining and confocal microscopy. Cell cultures were transiently transfected with wild-type EAAT1 or wild-type/mutant rASCT2 cDNAs, inserted into a modified pBK-CMV expression plasmid. Transfection were performed according to Jetprime transfection reagent and protocol (Polyplus). The cells were transfected after 24–36 hr then used for electrophysiological analysis.

### Electrophysiology

Currents associated with ASCT2 and EAAT1 were measured in the whole-cell current recording configuration. Whole-cell currents were recorded with an EPC7 patch-clamp Amplifier (ALA Scientific, Westbury, NY, USA) under voltage-clamp conditions. The resistance of the recording electrode was 3–6 M$\Omega$. Series resistance was not compensated because of the small whole-cell currents carried out by EAAT1, ASCTs. The composition of the solutions for measuring amino acid exchange currents in the anion conducting mode was: 140 mM NaMes (Mes = methanesulfonate), 2 mM MgGluconate$_2$, 2 mM CaMes$_2$, 10 mM 4-(2-hydroxyethyl)piperazine-1-ethanesulfonic acid (HEPES), with additional amino acid substrate (Glu or Ser) pH 7.3 (extracellular), and 130 mM NaSCN (SCN = thiocyanate, for EAAT1 NaSCN was replaced with KSCN), 2 mM MgGluconate$_2$, 5 mM ethylene glycol-bis(2-aminoethylether)-$N,N,N',N'$-tetraacetic acid (EGTA), 10 mM HEPES, pH 7.3 (intracellular), as published previously (*Wang et al., 2020*). For the measurement of the transport component of the current, intracellular SCN$^-$ was replaced with the Mes$^-$ anion.

### Voltage jump experiments

Voltage jumps (–100 to +60 mV) were applied to perturb the translocation equilibrium and to determine the voltage dependence of the anion conductance (*Wang et al., 2018*). To determine EAAT1/ASCT-specific currents, external solution contained 140 mM NaMes in the presence of varying concentrations of amino acid substrate. The internal solution contained 130 mM NaSCN (KSCN for EAAT1) in the presence of 10 mM amino acid substrate. Competitive blocker TBOA (*Shimamoto et al., 1998*; *Shimamoto et al., 2004*), or (R)-gamma-(4-biphenylmethyl)-L-proline (*Singh et al., 2017*) was used in control voltage jump experiments, yielding the unspecific current component, which was subtracted from the total current. Capacitive transient compensation and series resistance compensation of up to 80% was employed using the EPC-7 amplifier. Non-specific transient currents were subtracted in Clampfit software (Molecular Devices).

### Rapid solution exchange

Fast solution exchanges were performed using the SF-77B (Warner Instruments, LLC, MA, USA) piezo-based solution exchanger, allowing a time resolution in the 10 ms range. Amino acid substrate was applied through a theta capillary glass tubing (TG200-4, OD = 2.00 mm, ID = 1.40 mm. Warner Instruments, LLC, MA, USA), with the tip of the theta tubing pulled to a diameter of 350 µm and positioned at 0.5 mm to the cell (*Wang et al., 2022*; *Shi et al., 2021*). For paired-pulse experiments, currents were recorded with 10/20 ms interval time after removal of amino acid.

### Amino acid uptake

HEK293T cells were plated on collagen-coated 24-well plates ($0.5\times10^5$ cells/well) in Dulbecco's modified Eagle's medium containing 10% fetal bovine serum, penicillin/streptomycin (100 units/ml), 4.5 g/l glucose, sodium pyruvate, and glutamine (4 mM). Forty-eight hr post transfection with hASCT2 (see above), the cells were washed with uptake buffer (140 mM NaMes, 2 mM magnesium methanesulfonate, 2 mM calcium gluconate, 10 mM HEPES, pH 7.4, 5 mM glucose) two times and pre-incubated in the same buffer for 10 min. Subsequently, buffer was replaced with fresh buffer containing unlabeled

L-serine and 0.1 µCi of [$^{14}$C] L-serine (Moravek Biochemicals; total concentration 100 µCi/ml). After 5 min of incubation at room temperature, uptake was terminated by washing twice with 0.1 ml of uptake buffer on ice (after 5 min, uptake was in the linear range, as determined by quantifying the time dependence of uptake for times up to 5 min). The cells were then solubilized in 0.2 ml of 1% SDS, and radioactivity was measured by scintillation counting in 3 ml of scintillation fluid. Unspecific L-serine uptake was determined using the ASCT2 competitive inhibitor L-cis-BPE (*Garibsingh et al., 2021*).

## MD simulations

The model system for MD simulations was generated with VMD software (*Humphrey et al., 1996*) using an EAAT1 (5LLM) and ASCT2 (7BCS) structure. The final EAAT1/ASCT2 model was inserted in a pre-equilibrated POPC lipid bilayer with the dimensions about 130×130×90 Å. TIP3P water was added to generate a box measuring about 100 Å in the z-direction. NaCl were added at a total concentration of 0.15 M and the system was neutralized. The total numbers of atoms in the EAAT1 system were 168926, 193804 in ASCT2. Simulations were run using the CHARMM36 force field. NAMD (*Phillips et al., 2005*) simulations were performed using 2000 steps of minimization, followed by 10 ns equilibration runs under constant pressure conditions (NPT), and then for 100 ns, 300 ns or 500 ns (*Wang et al., 2021*). The RMSD (calculated from the peptide backbone) increased from 1.5 Å soon after simulation start to ~3 Å after 2 ns of equilibration, after which it was in steady state. The cutoff for local electrostatic and van der Waals interactions was set to 12 Å. For long-range electrostatic interactions, we used the particle-mesh Ewald method implemented in NAMD. Bonds to hydrogen atoms and TIP3P water were kept rigid using SHAKE. The time steps of the simulations were 2 fs. The time evolution of distances and distribution function analysis were calculated by Tcl/Tk programs built in the VMD software. For distance calculations, in EAAT1, we selected atoms Y127(CA) and M271(CA) for EAAT1 and (O) and (N) for UCPH-101 as reference atoms. In ASCT2$_{WT}$, we selected atoms F136(CZ) and I237(CD) for ASCT2$_{WT}$ and (N1) and (O2) for UCPH-101. In ASCT2$_{F136Y/I237M}$ distance calculation, we selected atoms Y136(OH) and M237(SD) from the transporter, and (N1) and (O2) from UCPH-101 as reference. We also selected S601(CA) and S353(CA) for ASCT2$_{WT}$ and ASCT2$_{F136Y/I237M}$ to evaluate the distance changes of substrate. The UCPH1-101 force field parameters were generated by SWISSPARAM. The parameter file is attached to the original data submitted with this manuscript.

## Data analysis

The data analysis was performed in Microsoft Excel and Microcal Origin software. Apparent inhibition constants ($K_i$) were obtained by fitting the dose-response curve of the current ($I$) with the following equation:

$$I = I_1 - I_2([UCPH - 101]/(K_i + [UCPH - 101]))  \tag{3}$$

Here, $I_1$ and $I_2$ are the current amplitudes in the absence of inhibitor, and at saturating UCPH-101 concentration. Error bars are shown as mean ± standard deviation, collected from recordings of 6–10 cells, for statistical analysis.

Transient signals of piezo-based solution-exchange results were analyzed in Clampfit software (Axon Instruments) by fitting with a sum of two exponential components. $I = I_1 \cdot \exp(-t/\tau_{rise}) + I_2 \cdot \exp(-t/\tau_{decay})$. Here, $I$ is the current amplitude, $\tau$ the time constant, and $t$ the time.

## Molecular docking

All docking calculations were conducted with the Schrödinger package (*Maestro, 2021*). ASCT2 (PDB ID: 7BCS) was prepared for docking using the Maestro Protein Preparation Wizard under default parameters. Glide v2020-2 was used to perform docking. The allosteric binding site was defined using the Maestro Receptor Grid Generation panel and the coordinates of the site were marked with UCPH-101, a reference ligand extracted from a superposition of an EAAT1 structure solved with bound UCPH-101 (PDB ID:5LLM or 5MJU generated the same result). We performed both rigid and flexible docking with and without hydrogen bond constraints at F377 and positional constraints in the non-mutated ASCT2 structure. However, docking poses were unrealistic and exhibited poor scores, and we thus, concluded they are unusable for further study.

## Docking UCPH-101 to mutated ASCT2

We remodeled the side chains of two residues in ASCT2 (i.e. F136Y and I273M) on a fixed backbone to mimic the UCPH-101 binding site in EAAT1, using SCWRL4 (*Krivov et al., 2009*). Grid preparation utilized positional constraints as highlighted in *Figure 1A* and excluded volume at F192 with a radius of 2.5 Å. The pose that had the best docking score was used to carry out the MD simulations.

## Virtual screening

The ZINC20 Lead-Like library ('In Stock'; 3.8 million compounds) was virtually screened against the WT ASCT2 allosteric site using Glide, as described above. The 1000 top-scoring compounds from the docking screen were further analyzed. In particular, because errors in docking can occur in large virtual screens, we inspected the docking poses of the top-ranking compounds to remove molecules with problematic pose or strained conformations, and prioritized them for experimental testing (*Bender et al., 2021*). We focused on molecules that interact with the conserved residues in the allosteric site (*Figure 2C*). We purchased 11 molecules for experimental testing.

## Acknowledgements

This study was supported by a grant from the National Institutes of Health (http://www.nih.gov. eresources.mssm.edu) (R01 GM108911) to AS and CG; and T32 CA078207 to RAG; and the R15 GM135843-01 awarded to CG.

## Additional information

### Funding

| Funder | Grant reference number | Author |
| --- | --- | --- |
| National Institutes of Health | R01 GM108911 | Avner Schlessinger Christof Grewer |
| National Institutes of Health | T32 CA078207 | Rachel-Ann Garibsingh |
| National Institutes of Health | R15 GM135843-01 | Christof Grewer |

The funders had no role in study design, data collection and interpretation, or the decision to submit the work for publication.

### Author contributions

Yang Dong, Data curation, Software, Formal analysis, Validation, Investigation, Visualization, Methodology, Writing - original draft, Writing - review and editing; Jiali Wang, Conceptualization, Resources, Data curation, Software, Formal analysis, Validation, Investigation, Visualization, Methodology, Writing - original draft, Writing - review and editing; Rachel-Ann Garibsingh, Software, Formal analysis, Validation, Investigation, Visualization, Methodology, Writing - review and editing; Keino Hutchinson, Data curation, Software, Formal analysis, Validation, Investigation, Visualization, Methodology, Writing - review and editing; Yueyue Shi, Data curation, Formal analysis, Validation, Investigation, Visualization, Methodology, Writing - review and editing; Gilad Eisenberg, Data curation, Formal analysis, Investigation; Xiaozhen Yu, Data curation, Formal analysis; Avner Schlessinger, Conceptualization, Data curation, Software, Formal analysis, Supervision, Funding acquisition, Validation, Investigation, Visualization, Methodology, Project administration, Writing - review and editing; Christof Grewer, Conceptualization, Data curation, Formal analysis, Supervision, Funding acquisition, Validation, Investigation, Visualization, Methodology, Writing - original draft, Project administration, Writing - review and editing

### Author ORCIDs

Jiali Wang http://orcid.org/0000-0002-9520-8140
Avner Schlessinger http://orcid.org/0000-0003-4007-7814
Christof Grewer http://orcid.org/0000-0002-8342-9878

Decision letter and Author response
Decision letter https://doi.org/10.7554/eLife.83464.sa1
Author response https://doi.org/10.7554/eLife.83464.sa2

## Additional files

### Supplementary files
- MDAR checklist
- Source code 1. UCPH-101 parameter file.
- Source code 2. UCPH-101 rtf file.

### Data availability
All data generated or analyzed during this study are included in the manuscript and supporting file. The original, source data files were uploaded as spreadsheets for Figures 3–10, Table 1, and the figure supplements for Figures 3–6 and 9. The MD parametrization files for UCPH-101 are also included.

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
