## [Editor Report]

The goal of this study is to identify allosteric modulators of an SLC-1 amino acid transporter, ASCT2, which has been implicated in cancer progression. By combining computational and docking methods with functional measurements, this study provides convincing evidence for a conserved allosteric SLC-1 inhibition mechanism. The findings are important to the fields of transporter mechanism and SLC-1 pharmacology.

---

## [Decision Letter]

**Decision letter after peer review:**

Thank you for submitting your article "Conserved allosteric inhibition mechanism in SLC1 transporters" for consideration by *eLife*. Your article has been reviewed by 3 peer reviewers, including Randy B Stockbridge as Reviewing Editor and Reviewer #1, and the evaluation has been overseen by Richard Aldrich as the Senior Editor.

Essential revisions:

The following revisions are essential to support the central claims of the manuscript regarding the conservation of an allosteric inhibition site in SLC1 transporters:

1. The MD simulations should include detailed convergence analyses to support the conclusions. According the X-ray crystal structure, the UCPH-101 binding pocket is only accessible via the lipid membrane. It seems unlikely that the 100-ns timescales of the simulations is sufficiently long to permit conclusions to be drawn. The minimum length of the MD replicas should be chosen such that complete dissociation of UCPH-101 is observed in all the ASCT2 replicas. The authors should also comment on how independence of the simulation replicas is ensured.

2. A major conclusion of the paper is that ASCT2 and EAAT1 share a conserved allosteric inhibition mechanism. However, the reviewers are not convinced that an allosteric inhibition mechanism has been established for ASCT2. The data for UPC101 suggests a mixed inhibition mechanism (lines 460-465 and Figure 7). The authors should comment on why the mechanism of UPC101 inhibition differs for the two SLC1 subtypes. Likewise, the evidence that #302 acts as an allosteric inhibitor (i.e., that the inhibition constant is independent of substrate concentration) is quite sparse, since inhibition was only tested at two substrate concentrations, and different inhibitor concentrations were used at these two substrate concentrations. An experiment with constant #302 concentration and varying substrate concentration should be performed to establish the allosteric inhibition mechanism.

3. The only evidence that #302 binds to the same binding pocket as UPC101 is from docking studies. The authors should provide data that #302 binds in the same pocket in ASCT-2 as UCPH-101, for example by testing the same point mutations that were used for the UCPH-101 experiments.

4. The manuscript is written in a very technical manner. The reviewers recommend that the authors revise the manuscript for readability, taking into consideration *eLife*'s broad readership. In addition, the studies of UPC101 inhibition of EAAT1 largely confirm previous studies that established the allosteric inhibition mechanism (PMID: 35192345; PMID: 33597752). In the text, the authors should clearly explain how their studies of UPC101/EAAT1 inhibition are significant advances beyond what is already known about the inhibition mechanism, or consider shifting the emphasis of the paper towards the more novel studies involving ASCT2 mutagenesis and inhibition.

*Reviewer #2 (Recommendations for the authors):*

The manuscript is written in a very technical way and should be edited to improve readership by a broader *eLife* community.

Throughout, 'trimerization' and 'scaffold' domain are used interchangeably. One term to describe this domain should be selected and used throughout the manuscript.

Line 24: the EAAT1 structure was solved by X-ray crystallography, not cryo-EM.

Line 53: the counter-transport of K^+^ and co-transport of H^+^ that also occur with glutamate transport are not mentioned.

Line 63-64: the description of the TMD making up the transport domain and the scaffold domain is incorrect.

Figure 1. in panel B, TFB-TBOA and UCPH-101 should be indicated clearly with labels and different colors. Line 233 states that UCHP-101 binding pocket is over 15A away from the substrate and cation bindings sites, but these are not indicated in the figure and it is not correct that the compound is 15A away from the substrate and all three of the Na^+^ ion binding sites.

What is the compound bound in Figure 2A? it would be useful to indicate the membrane in all structure figures as well as 'outside' and 'inside' the cell to orient the reader.

Figure 3, panel B – it is difficult to see the data in this panel, a logarithmic scale would be better.

Line 358: the word significantly is used but I can see no statistical analysis and the effect does not seem overly large.

The section (lines 362-379) is confusing, and I am not sure what these experiments are trying to show, the data in Figure S4 does not look different at all with error bars overlapping.

Table 1 shows the Km for various substrates for the ASCT2 mutants. There is no discussion or explanation offered as to why residues distant from the substrate binding site are impacting affinity and selectivity.

Figure 7D. Lines 460-465 state that the Ki for the single mutant does change as a function of serine concentration, and a mixed mode of inhibition is suggested. Why would the mode of inhibition change if UCPH-101 is binding in the same site? In addition, from the data in Figure 7D it appears to this reviewer that the Ki for UCPH-101 is also changing for the double mutant. This needs to be clarified.

Did the authors test if UCPH-101 inhibits the more closely related ASCT1? What is the conservation like in the UCPH-101 binding site?

*Reviewer #3 (Recommendations for the authors):*

(1) I strongly recommend to include experimental evidence that (i) compound #302 also mediates allosteric inhibition (i.e., that the inhibition constant is independent of substrate concentration) and (ii) that it binds to the same binding pocket in ASCT-2 as UCPH-101 (perhaps using the same point mutations that you have used for the UCPH-101 experiments).

(2) The MD simulations should include detailed convergence analyses to support their conclusions. Since the UCPH-101 binding pocket is presumably (according the X-ray crystal structure) only accessible via the lipid membrane, I doubt that the 100-ns timescales of your simulations are sufficiently long to permit reasonable conclusions. In any case, I would recommend that the minimum length of your MD replicas should be chosen such that you observe complete dissociation of UCPH-101 in all the ASCT2 replicas.

Furthermore, Figure 5 suggests you conducted four simulation replicas for each condition, whereas the main text says you have six replicas (line 388), what is correct?

Finally, how did you ensure independence of your simulation replicas?

(3) The conclusions of your MD simulations will also crucially depend on the chemical accuracy of your UCPH-101 model. Unfortunately, I could not find how you parameterized this compound consistent with the CHARMM force field. Please provide this information.

---

## [Author Response]

Reviewer #2 (Recommendations for the authors):The manuscript is written in a very technical way and should be edited to improve readership by a broader eLife community.

We agree. To fit with the broader readership of *eLife* (also see Editor's comment), we have completely revised the section of the manuscript on the mechanism of UCPH-101 interaction with EAAT1, removing the overly-technical detail, in particular the caged glutamate experiments (previous Figure S3), which do not contribute strongly to the general conclusions. We also revised the Discussion section accordingly. For the ASCT2 Results section, however, we believe that a thorough description of the anion-current vs transport-current recording experiments, as well as the paired-pulse analysis, are crucial for the non-expert reader to be able to follow the interpretation of the data. Therefore, we made less substantial changes to this section.

Throughout, 'trimerization' and 'scaffold' domain are used interchangeably. One term to describe this domain should be selected and used throughout the manuscript.

We have replaced "scaffold" with "trimerization" in the revised manuscript.

Line 24: the EAAT1 structure was solved by X-ray crystallography, not cryo-EM.

Thanks for this correction, this is now fixed.

Line 53: the counter-transport of K^+^ and co-transport of H^+^ that also occur with glutamate transport are not mentioned.

These are now mentioned and the relevant references are cited.

Line 63-64: the description of the TMD making up the transport domain and the scaffold domain is incorrect.

Thanks for spotting this, this is now fixed.

Figure 1. in panel B, TFB-TBOA and UCPH-101 should be indicated clearly with labels and different colors. Line 233 states that UCHP-101 binding pocket is over 15A away from the substrate and cation bindings sites, but these are not indicated in the figure and it is not correct that the compound is 15A away from the substrate and all three of the Na^+^ ion binding sites.What is the compound bound in Figure 2A? it would be useful to indicate the membrane in all structure figures as well as 'outside' and 'inside' the cell to orient the reader.

We thank the reviewer for this comment. We removed the Na^+^ from the sentence and labeled the figures as suggested by the reviewer.

Figure 3, panel B – it is difficult to see the data in this panel, a logarithmic scale would be better.

We agree, a logarithmic scale would be preferable. We have revised Figure 3B accordingly.

Line 358: the word significantly is used but I can see no statistical analysis and the effect does not seem overly large.

This figure, and its description in the text, have been removed.

The section (lines 362-379) is confusing, and I am not sure what these experiments are trying to show, the data in Figure S4 does not look different at all with error bars overlapping.

Yes, the data in former Figure S4 (now Figure 4—figure supplement 1) are not significantly different, showing that bound UCPH-101 does not interfere with Na^+^ binding to the apo-form of the transporter. We have revised this paragraph to make this point more clear.

Table 1 shows the Km for various substrates for the ASCT2 mutants. There is no discussion or explanation offered as to why residues distant from the substrate binding site are impacting affinity and selectivity.

We performed these measurements as controls, to check whether the mutant transporters preserved WT-like activity. For serine as the substrate, this is clearly the case. However, for alanine and glutamine, some deviations from this expectation were noted. The reason for this is not clear, because these residues are far from the substrate binding site, as the reviewer points out. We can speculate that there are some indirect effects of these mutations on ASCT2 function that are not related to their contribution to the proposed allosteric site. We added a sentence explaining this in the revised manuscript.

Figure 7D. Lines 460-465 state that the Ki for the single mutant does change as a function of serine concentration, and a mixed mode of inhibition is suggested. Why would the mode of inhibition change if UCPH-101 is binding in the same site? In addition, from the data in Figure 7D it appears to this reviewer that the Ki for UCPH-101 is also changing for the double mutant. This needs to be clarified.

Yes, this is correct, the [serine] dependence of the K_i_ for UCPH-101 is consistent with an inhibition mechanism, in which the compound binds somewhat more strongly to the mutant transporters in the absence of serine than in the presence, in other words a mixed mechanism. We have now revised this sentence, see also response to Reviewer #1, who also raised this point.

Did the authors test if UCPH-101 inhibits the more closely related ASCT1? What is the conservation like in the UCPH-101 binding site?

This is a great point. Yes, UCPH101 also inhibits ASCT1 function. However, we feel that detailed analysis of the effect on ASCT1 is beyond the scope of this manuscript. Instead, we also tested our new compound, #302, with ASCT1 and it does, in fact, inhibit ASCT1 substrate-induced currents (new Figure 10J). This is expected, since the predicted allosteric binding site is largely conserved between ASCTs 1 and 2 (see revised sequence alignment in the supplement, which now includes ASCT1).

Reviewer #3 (Recommendations for the authors):(1) I strongly recommend to include experimental evidence that (i) compound #302 also mediates allosteric inhibition (i.e., that the inhibition constant is independent of substrate concentration) and (ii) that it binds to the same binding pocket in ASCT-2 as UCPH-101 (perhaps using the same point mutations that you have used for the UCPH-101 experiments).

We agree. These experiments are now included in the revised version of the manuscript (new Figure G and I).

(2) The MD simulations should include detailed convergence analyses to support their conclusions. Since the UCPH-101 binding pocket is presumably (according the X-ray crystal structure) only accessible via the lipid membrane, I doubt that the 100-ns timescales of your simulations are sufficiently long to permit reasonable conclusions. In any case, I would recommend that the minimum length of your MD replicas should be chosen such that you observe complete dissociation of UCPH-101 in all the ASCT2 replicas.

We have extended the length of the MD simulations for ASCT2 wild-type and double mutant transporter to at least 300 ns (some to 500 ns, new Figure 5, figure supplement 1). Even with a 5-fold extended time scale, we observed only two UCPH dissociation events for ASCT2_WT_ (total of six trajectories, dissociation occurred into the lipid bilayer), and none for the double mutant transporter. In our view, this points to the occluded nature of UCPH-101 in the binding site of the transporter, resulting in slow dissociation kinetics, consistent with experimental results. Unfortunately, due to time restrictions, we were not able to run these simulations at an even longer time scale.

Furthermore, Figure 5 suggests you conducted four simulation replicas for each condition, whereas the main text says you have six replicas (line 388), what is correct?Finally, how did you ensure independence of your simulation replicas?

We performed six replicas, but only four are shown in Figure 5. This is now explained in the main text.

(3) The conclusions of your MD simulations will also crucially depend on the chemical accuracy of your UCPH-101 model. Unfortunately, I could not find how you parameterized this compound consistent with the CHARMM force field. Please provide this information.

We apologize for the omission of this critical point. The information on the parametrization of UCPH101 is now provided in the original data (deposited parameter and topology files). We did not use physical chemistry experimentation to validate UCPH-101 MD parameters. Since the MS results are supportive in nature, and not critical to the overall conclusions, we feel that experimental validation is beyond the scope for this manuscript.